# Impacts of Dietary Protein and Niacin Deficiency on Reproduction Performance, Body Growth, and Gut Microbiota of Female Hamsters (*Tscherskia triton*) and Their Offspring

Jidong Zhao,[a,b] Wei Lu,[a] Shuli Huang,[a] Yvon Le Maho,[c,d] Caroline Habold,[c] ⓘ Zhibin Zhang[a,e]

[a]State Key Laboratory of Integrated Management of Pest Insects and Rodents, Institute of Zoology, Chinese Academy of Sciences, Beijing, People's Republic of China
[b]Shaanxi Key Laboratory of Qinling Ecological Security, Shaanxi Institute of Zoology, Xi'an, People's Republic of China
[c]University of Strasbourg, CNRS, IPHC, UMR 7178, Strasbourg, France
[d]Scientific Centre of Monaco, Monaco Principality, Monaco
[e]CAS Center for Excellence in Biotic Interactions, University of Chinese Academy of Sciences, Beijing, People's Republic of China

**ABSTRACT** Food resources are vital for animals to survive, and gut microbiota play an essential role in transferring nutritional materials into functional metabolites for hosts. Although the fact that diet affects host microbiota is well known, its impacts on offspring remain unclear. In this study, we assessed the effects of low-protein and niacin-deficient diets on reproduction performance, body growth, and gut microbiota of greater long-tailed hamsters (*Tscherskia triton*) under laboratory conditions. We found that maternal low-protein diet (not niacin deficiency) had a significant negative effect on reproduction performance of female hamsters (longer mating latency with males and smaller litter size) and body growth (lower body weight) of both female hamsters and their offspring. Both protein- and niacin-deficient diets showed significant maternal effects on the microbial community in the offspring. A maternal low-protein diet (not niacin deficiency) significantly reduced the abundance of major bacterial taxa producing short-chain fatty acids, increased the abundance of probiotic taxa, and altered microbial function in the offspring. The negative effects of maternal nutritional deficiency on gut microbiota are more pronounced in the protein group than the niacin group and in offspring more than in female hamsters. Our results suggest that a low-protein diet could alter gut microbiota in animals, which may result in negative impacts on their fitness. It is necessary to conduct further analysis to reveal the roles of nutrition, as well as its interaction with gut microbes, in affecting fitness of greater long-tailed hamsters under field conditions.

**IMPORTANCE** Gut microbes are known to be essential for hosts to digest food and absorb nutrients. Currently, it is still unclear how maternal nutrient deficiency affects the fitness of animals by its effect on gut microbes. Here, we evaluated the effects of protein- and niacin-deficient diets on mating behavior, reproduction, body growth, and gut microbiota of both mothers and offspring of the greater long-tailed hamster (*Tscherskia triton*) under laboratory conditions. We found that a low-protein diet significantly reduced maternal reproduction performance and body growth of both mothers and their offspring. Both protein and niacin deficiencies showed significant maternal effects on the microbial community of the offspring. Our results hint that nutritional deficiency may be a potential factor in causing the observed sustained population decline of the greater long-tailed hamsters due to intensified monoculture in the North China Plain, and this needs further field investigation.

**KEYWORDS** cropland monoculture, greater long-tailed hamster, gut microbiota, low-protein diet, niacin deficiency

**Ad Hoc Peer Reviewer** Dominik Schmid

Address correspondence to Zhibin Zhang, zhangzb@ioz.ac.cn.
The authors declare no conflict of interest.

Anutritionally balanced diet is vital for maintaining biological processes and the survival of all organisms. Nutritional deficiency can reduce the fitness of animals by disrupting behavior and prohibiting growth or reproduction. Amino acids are essential for protein biosynthesis and regulation of cell signaling and metabolic pathways. Low protein intake has been shown to be associated with increased inflammation risk (1), prohibition of postnatal body growth (2), and reduction of fertility of adult females (3) and offspring (4) in many murine species. Niacin (i.e., vitamin $B_3$, or nicotinamide) is one of the water-soluble vitamins that are involved in various biological functions for maintenance of normal growth and reproduction of animals. Deficiency in dietary niacin may disrupt the maternal behavior of hamsters (3) and increase risk of disease in humans, such as pellagra and diarrhea (5) or colitis (6). Although either a low-protein diet (LPD) (2, 7–9) or a niacin-deficient (niacin⁻) diet (10–12) can cause abnormality in reproduction of female animals, the interaction of dietary protein and niacin deficiency has not been fully investigated.

Nutrient deficiency may be one of potential reasons for population decline in some wild animal species. Intensified monoculture is suggested to be one of the key factors that threatens rodent species in cropland (13–15). Crop monoculture practices may influence the diet structure of cropland wildlife by reducing overall biodiversity in the habitat (3, 15, 16). For instance, the European hamster was once widely distributed in Eurasian states, but has now become endangered worldwide. Deficiencies in dietary protein and niacin, resulting from wheat and maize monoculture, may have contributed to the reproductive failure of European hamsters, thus accelerating the decline of their wild population (3). However, the maternal effects of protein and niacin deficiency on offspring due to gut microbial imbalances remain unclear.

Gut microbiota are essential in maintaining normal function and health of hosts: e.g., food digestion, immune response, metabolic homeostasis, and other bodily processes (17–20). Access to food resources is a key factor in shaping homeostasis of the gut microbial community in animals (21, 22), while disturbances in gut microbes may detrimentally affect gut metabolic function (18, 23–25). Nutrient deficiency may disrupt gut microbiota, also reducing fitness of hosts and their offspring. Dietary protein deficiency can alter the microbial community (26–29), but available results are often contradictory (28). Niacin deficiency can also affect gut microbes, as well as the health of both hosts and their offspring animals (10–12, 30). Maternal gut microbiota may affect offspring during vaginal delivery and breastfeeding (31, 32), but the effects of maternal nutrient deficiency on offspring need further investigation.

The greater long-tailed hamster (*Tscherskia triton*) is widely distributed in north China and some parts of Russia and North and South Korea. It was previously a predominant pest rodent species in the farmlands of the North and Northeast China Plain (33). Recently, sustained population decline of this species has been reported (34–36). A sympatric species, the Chinese striped hamster (*Cricetulus barabensis*), is also suffering from a similar sustained population decline (37). From 1981 to 2015, areas in the North China Plain used for planting soybeans and cotton decreased, while areas for maize increased (38). Similar to the case of European hamsters (3), dietary changes resulting from planting a monoculture may have contributed to the observed population decline of *T. triton*, but this hypothesis has not been tested.

The purpose of this study was to examine the effects of maternal protein and niacin deficiency on body growth, reproduction performance, and gut microbiota of the greater long-tailed hamsters under laboratory conditions. We designed a 2- by 2-factor experiment, making up four diet treatment groups with combinations of low-protein diet (LPD) or normal-protein diet (NPD) and niacin-supplemented (niacin⁺) or niacin-deficient (niacin⁻) diet for female hamsters: (i) the NPD-niacin⁺ group, (ii) LPD-niacin⁺ group, (iii) NPD-niacin⁻ group, and (iv) LPD-niacin⁻ group (for details, see Materials and Methods and Fig. 1). Adult female hamsters were fed with standard and modified AIN93G rodent chow containing the above four different diets for 1 month; body weight was measured, and feces were collected in order to analyze gut microbes. Then, the females were assigned to cohabit with normal adult males for

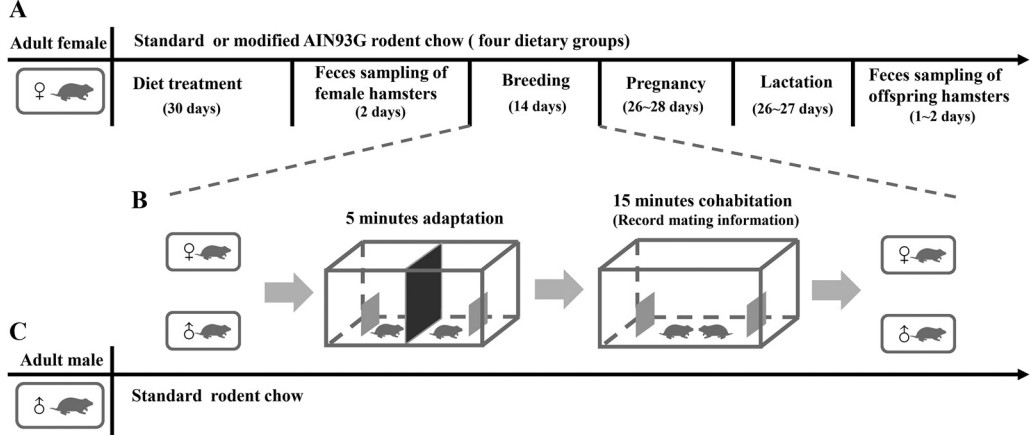

**FIG 1** Schematic overview of the experimental design. (A and C) Experimental time course for (A) adult females (mother hamsters) and (C) adult males (for mates of female hamsters) of greater long-tailed hamsters. Adult female hamsters ($n = 51$) were fed with standard and modified AIN93G rodent chow of four different diets in Table 2, and adult males ($n = 51$) were fed with only standard AIN93G rodent chow. (B) Cohabitation tests for breeding and measuring the mating behaviors of adult female hamsters. For details, see Materials and Methods.

14 days. During the cohabitation tests (each test lasted 1 day), we measured the mating behaviors of the female hamsters, including mount latency of the female (time needed for a female to have the first successful mating with a male), mount frequency (number of copulations within 15 min after cohabitation with an adult male), and number of cohabitation tests (number of cohabitation tests needed for a maternal female to successfully mate with an adult male). After successful mating, females were housed individually until giving birth, and their litter sizes were measured, body weights of pups at weaning were measured, and feces were collected to analyze gut microbes. The effects of maternal protein and niacin deficiency on body growth, reproduction performance, and gut microbiota of the greater long-tailed hamsters were statistically tested. We focused on testing for the following two hypotheses: (i) maternal low protein and niacin deficiency would have a negative effect on body growth and reproduction performance of female hamsters and the body growth of offspring, and (ii) maternal low protein and niacin deficiency would alter gut microbes of both maternal and offspring hamsters. If these hypotheses held, it would provide us with important implications that nutritional deficiency may play a potential role in population decline of *T. triton* under the intensified monoculture in the North China Plain, and more work needs to be done to confirm this potential issue.

## RESULTS

**Effects of maternal protein- and niacin-deficient diets on body weight of female hamsters and their offspring.** We found there was no overall significant effect of protein ($F = 2.56$; $P = 0.116$)- and niacin ($F = 0.82$; $P = 0.371$)-deficient diets on body weight of female hamsters during the study period (see Table S2 in the supplemental material). We found there was a significant interaction effect between protein diet and sampling time ($F = 25.41$; $P < 0.0001$) (Fig. 2A); the body weight of female hamsters in the maternal low-protein diet (LPD) groups was significantly lower than that in the maternal normal-protein diet (NPD) groups at days 14, 18, 22, and 30 (Fig. 2A). Maternal LPD significantly decreased offspring body weight during the weaning period ($P < 0.0001$) (Fig. 2B and Table S2). We did not find a significant effect of niacin deficiency on body weight for both female hamsters and offspring ($P > 0.05$) (Table S2).

**Effects of protein- and niacin-deficient diets on reproduction of female hamsters.** We found maternal LPD significantly increased the number of cohabitation tests (one test per day) needed for a female to successfully mate with an adult male compared to NPD ($P < 0.001$) (Fig. 2C and Table S3). There was no such significant effect of niacin deficiency ($P < 0.05$) (Table S3). We found maternal LPD significantly increased the mount latency of females compared to that in the NPD group ($P < 0.001$) (Fig. 2D

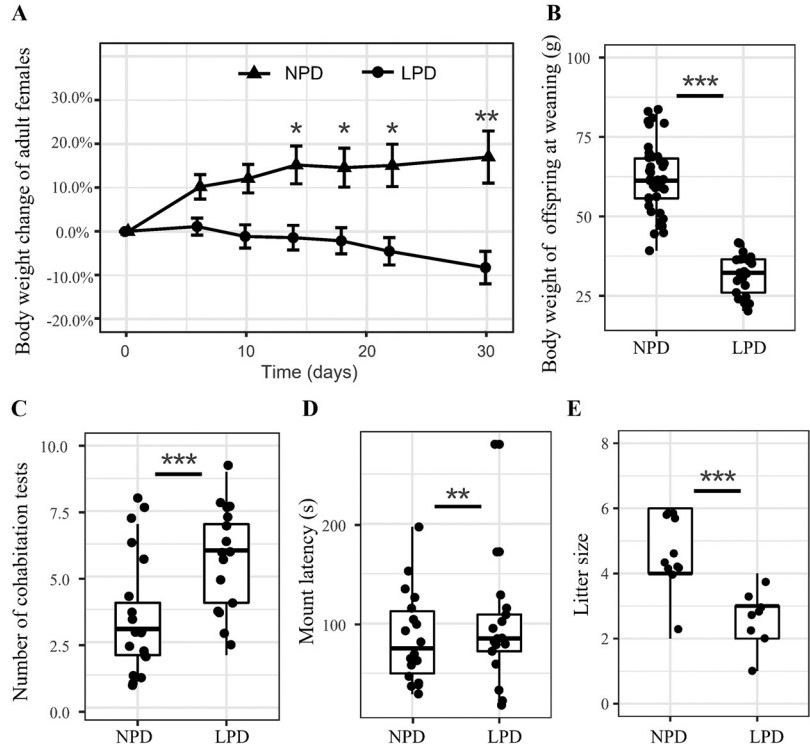

**FIG 2** Significant effects of protein diet on body weight, mating behaviors, and reproduction of female hamsters and body weight of offspring hamsters. (A) Body weight change (mean ± standard error of the mean [SEM]) of female hamsters under normal-protein diet (NPD) and low-protein diet (LPD); (B) body weight of offspring hamsters at weaning period; (C) number of cohabitation tests for successful mating of a female hamster with a male hamster; (D) mount latency of female hamsters at their first copulation; (E) litter size of female hamsters. *, $P < 0.05$; **, $P < 0.01$; ***, $P < 0.001$.

and Table S3). However, there was no significant difference in mount frequencies of female hamsters with NPD and LPD ($P > 0.05$; NPD, 6.41 ± 0.49; LPD, 7.11 ± 0.54) or niacin+ and niacin− ($P > 0.05$; niacin+, 7.07 ± 0.67; niacin−, 6.55 ± 0.41) (Table S3). An LPD significantly decreased litter size in female hamsters ($P < 0.001$) (Fig. 2E and Table S3). There was no significant difference in sex ratio (male/female) in offspring with NPD and LPD ($P > 0.05$; NPD, 2.19; LPD, 1.63) or niacin+ and niacin− ($P > 0.05$; niacin+, 2.64; niacin−, 1.46) (Table S3). We observed one case of infanticide in a maternal female in the niacin− diet group (Table S1).

**Effects of protein- and niacin-deficient diets on gut microbiota of female hamsters and their offspring.** We found a significant effect of protein-deficient diet on the gut microbial community of female hamsters during the 1-month adaptation ($P = 0.002$) (Table 1 and Fig. 3A), but there was no such significant effect of a niacin-deficient diet on female hamsters ($P = 0.363$) (Table 2 and Fig. 3A). A maternal protein-deficient diet significantly altered the gut microbial community of offspring at weaning ($P = 0.001$) (Table 1 and Fig. 3B), while the niacin-deficient diet failed to affect the overall gut microbial community of offspring hamsters ($P = 0.207$) (Table 1 and Fig. 3B). β dispersion analysis indicated maternal LPD significantly reduced interindividual variation in the gut microbial community of offspring hamsters compared to the maternal NDP diet ($P = 0.001$) (Fig. 3B and Table 1).

We found no significant effect ($P > 0.05$) of protein-deficient or niacin-deficient diets on α diversity indices of female hamsters (Table S4). However, we found that maternal LPD significantly decreased the Shannon diversity, Chao1 index, and observed features in offspring hamsters (Fig. 3C and Table S4), while the maternal niacin-deficient diet significantly increased the Shannon diversity, Chao1 index, and observed features in offspring hamsters (Fig. 3D and Table S4). Both maternal LPD ($P = 0.500$) and maternal niacin− ($P = 0.543$) had no significant effect on the phylogenetic diversity index of gut microbiota in hamster

**TABLE 1** Effects of protein and niacin diets on interindividual variation and composition of gut microbiota of female and offspring hamsters[a]

| Parameter | Female hamster | | | Offspring hamster | | |
|---|---|---|---|---|---|---|
| | F or pseudo-$F$[b] | P value | $R^2$ | F or pseudo-$F$[b] | P value | $R^2$ |
| $\beta$ dispersion | | | | | | |
| Protein (NPD vs LPD) | 0.43 | 0.503 | | 25.03 | 0.001 | |
| Niacin (niacin$^+$ vs niacin$^-$) | 0.0012 | 0.974 | | 0.90 | 0.359 | |
| | | | | | | |
| PERMANOVA | | | | | | |
| Protein (NPD vs LPD) | 1.87 | 0.002 | 0.052 | 2.68 | 0.001 | 0.081 |
| Niacin (niacin$^+$ vs niacin$^-$) | 1.04 | 0.363 | 0.029 | 1.11 | 0.207 | 0.039 |
| Protein × niacin | 1.09 | 0.262 | 0.030 | 0.97 | 0.584 | 0.027 |
| Dam's ID | | | | 3.80 | 0.001 | 0.439 |

[a]NPD, normal-protein diet; LPD, low-protein diet; niacin$^+$, niacin-supplemented diet; niacin$^-$, niacin-deficient diet.
[b]F values are shown for $\beta$ dispersion parameters, and pseudo-F values are shown for PERMANOVA parameters.

offspring (Table S4). Sex and litter size showed nonsignificant effects on $\alpha$ diversity in the offspring (all P values of >0.05). Fecal microbiota of both female hamsters and their offspring consisted of nine predominant phyla, including *Firmicutes*, *Bacteroidetes*, TM7, *Proteobacteria*, *Verrucomicrobia*, *Actinobacteria*, *Tenericutes*, *Spirochaetes*, and *Cyanobacteria* (Fig. 4).

Analysis with ANCOMBC (analysis of compositions of microbiomes with bias correction) indicated that there was no significant differential taxa at the phylum level in female hamsters between protein- and niacin-deficient diets (adjusted $P > 0.05$). However, we found the absolute abundance of *Proteobacteria* (adjusted $P < 0.0001$) and *Spirochaetes* (adjusted $P < 0.0001$) was significantly decreased, while that of *Actinobaceria* (adjusted $P < 0.0001$) in the offspring was significantly increased in the maternal LPD group. There was no significant such difference of phyla (adjusted $P > 0.05$) between the maternal niacin$^+$ and niacin$^-$ diet groups. A protein diet significantly altered the absolute abundance of 27 genera from *Firmicutes*, *Bacteroidetes*, *Proteobacteria*, *Spirochaetes*, *Actinobacteria*, *Tenericutes*, *Deferribacteres*, and *Fusobacteria* of female hamsters (Fig. 5A and Table S5) and 41 genera from *Firmicutes*, *Bacteroidetes*, *Proteobacteria*, *Spirochaetes*, *Actinobacteria*, *Tenericutes*, *Cyanobacteria*, and *Deferribacteres* in offspring (Fig. 5C and Table S6). Maternal LPD significantly decreased the absolute abundance of the genera *Dorea*, *Desulfovibrio*, *Helicobacter*, *Sporobacter*, *Treponema*, and *Ruminococcus* but increased the absolute abundance of *Lactobacillus* and *Bifidobacteria* in gut microbiota of the offspring (Fig. 5C and Table S6). Absolute abundance of the genera *Anaeroplasma*, *Gemmiger*, and *Spirochaeta* showed the same response in both female and offspring hamsters to the LPD (Fig. 5A and C and Tables S5 and S6). Also, niacin$^-$ significantly decreased the absolute abundance of the genera *Macrococcus*, *Mucispirillum*, *Clostridium*, *Arthrobacter*, *Alkanindiges*, and *Hespellia*, but it increased the abundance of the genera *Spirochaeta* and *Proteiniborus* (Fig. 5B and Table S7).

Analysis using the Tax4Fun2 algorithm indicated that neither a protein- nor niacin-deficient diet significantly altered the microbial functional orthologs (NPD versus LPD, $P > 0.05$; niacin$^+$ versus niacin$^-$, $P > 0.05$) (Table S8) and KEGG pathways (NPD versus LPD, $P > 0.05$; niacin$^+$ versus niacin$^-$, $P > 0.05$) (Table S8) in female hamsters. However, microbial functional orthologs in offspring were significantly changed by maternal LPD ($P = 0.021$) (Fig. 6 and Table S8), but the KEGG pathway was not significantly affected ($P = 0.073$). We found no significant difference in functional orthologs ($P > 0.05$) or KEGG pathways ($P > 0.05$) between niacin diet groups (Table S8). LEfSe (linear discriminant analysis [LDA] effect size) analysis revealed 53 enriched functional orthologs and 32 enriched pathways (KEGG level 1) in offspring hamsters (Fig. 6 and Tables S9 and S10), and most of the altered functions and pathways are associated with protein metabolism.

## DISCUSSION

It is known that protein or niacin deficiencies may negatively affect the body growth and reproduction of animals, but their maternal impacts on fitness and gut microbiota of

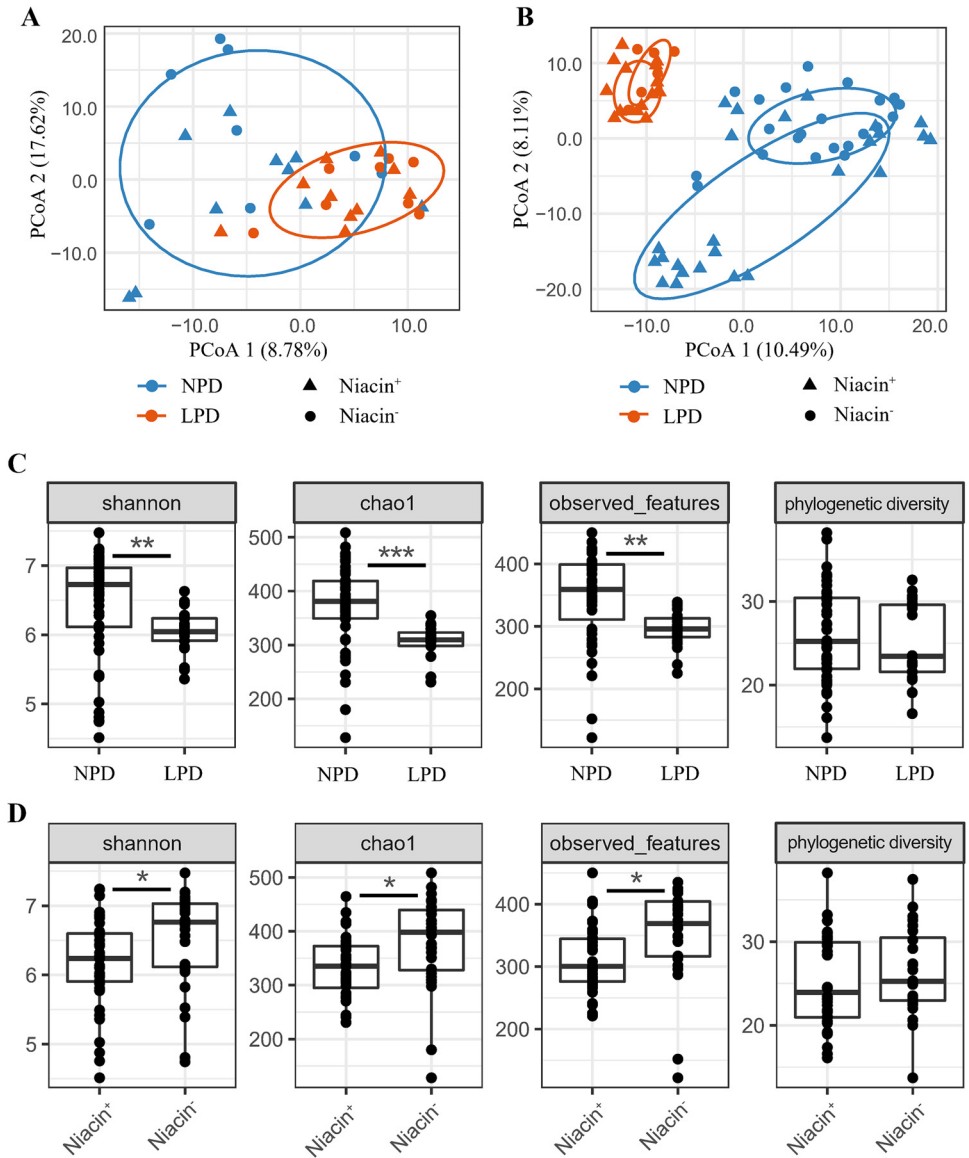

**FIG 3** Differences in gut microbial diversity of hamsters under different protein diets and niacin diets. (A and B) Aitchison distance PCoA for female hamsters (A) and offspring hamsters (B). (C and D) $\alpha$ diversity indices of offspring for protein diet groups (C) and niacin diet groups (D). *, $P < 0.05$; **, $P < 0.01$; ***, $P < 0.001$. Abbreviations: NPD, normal-protein diet; LPD, low-protein diet; Niacin+, niacin-supplemented diet; Niacin−, niacin-deficient diet.

offspring are not fully investigated. In this study, we found maternal LPD, but not niacin− diet, significantly and negatively affected mating behavior, reproduction, and body growth of maternal greater long-tailed hamsters (*T. triton*) and body growth of their offspring. However, both protein and niacin deficiencies showed significant maternal effects on the microbial community of their offspring. Maternal LPD (not niacin− diet) significantly altered the microbial functions of offspring. Our results generally support the two hypotheses, with additional findings that protein deficiency has a more pronounced effect than niacin deficiency and the malnutrition effect is greater in offspring than in female hamsters.

**Effects of nutritional deficiency on hamster body growth.** Administration of a nutritional deficiency diet during pregnancy and lactation can decrease body weight of both maternal females and their offspring. A maternal LPD may restrict postnatal body growth (2), promote inflammation risk (1), and impair the fertility of offspring in later life (4) in laboratory rodent species. Niacin supplements may attenuate the weight loss in rats

**TABLE 2** Nutritional parameters in four protein and niacin diet groups for female hamsters in a 2- by 2-factor experiment[a]

| Nutritional parameter | Result for diet group: | | | |
|---|---|---|---|---|
| | NPD-niacin[+] | NPD-niacin[−] | LPD-niacin[+] | LPD-niacin[−] |
| Protein, % | **22.2** | **22.2** | **8.8** | **8.8** |
| Fat, % | 6.8 | 6.8 | 6.9 | 6.9 |
| Niacin, mg/kg | **30.0** | **0** | **30.0** | **0** |
| Carbohydrate, % | 55.5 | 55.5 | 67.5 | 67.5 |
| Total energy (kcal/kg) | 3.72 | 3.72 | 3.68 | 3.68 |

[a]The diet formula was modified based on standard AIN93G rodent chow. Standard (basal) AIN93G rodent chow contained normal protein with niacin supplement. Abbreviations: NPD, normal-protein diet; LPD, low-protein diet; niacin[+], niacin-supplemented diet; niacin[−], niacin-deficient diet. Differences in nutrients are highlighted in boldface.

with severe colitis (6). In this study, we found a maternal LPD significantly reduced body weight of offspring at weaning, with an almost 50% decrease in body weight, compared to the niacin-deficient diet (Fig. 2B), which is consistent with previous findings (2, 7, 8). The effects of niacin deficiency on body weight were not observed in our study; however, we found one case of maternal infanticide in the niacin-deficient group, which is similar to the finding in European hamsters fed a maize-based diet (3). It is not clear whether the observed infanticide behavior was associated with niacin deficiency or not in our study due to the observation of only a single case. Decreased body weight at weaning is likely caused by a decrease in milk production or composition change (9). However, the roles of milk production or its composition change in offspring are not clear because we did not have these data. Because the body weight of female hamsters was also reduced in LPD groups, it is likely that protein levels were also decreased in maternal milk. In future studies, it is necessary to examine the milk production and nutritional changes in maternal animals and identify how these changes affect their offspring.

**Effects of nutritional deficiency on reproduction performance of hamsters.** Maternal nutritional deficiency may negatively affect reproduction performance of animals. An LPD may induce cognition decline and anxiety-like behavior in mice (39). Litter size was significantly affected by an LPD in our study, which is a result similar to

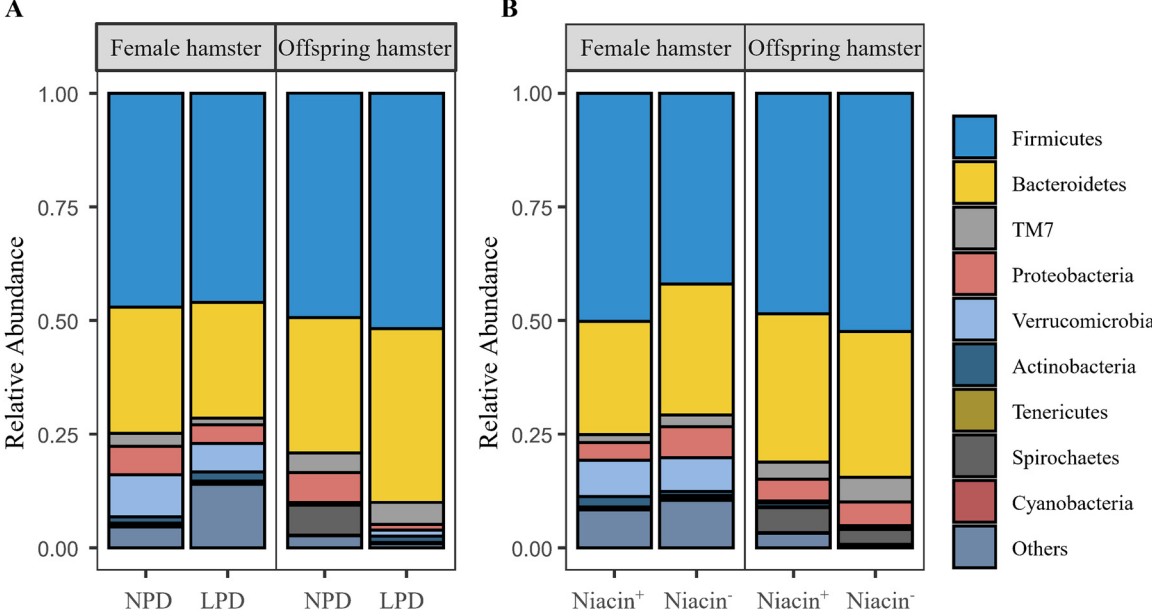

**FIG 4** Difference of the relative abundances of the dominant gut microbial phyla in female and offspring hamsters between different protein diet (A) and niacin diet (B) groups at the phylum level. Abbreviations: NPD, normal-protein diet; LPD, low-protein diet; Niacin[+], niacin-supplemented diet; Niacin[−], niacin-deficient diet.

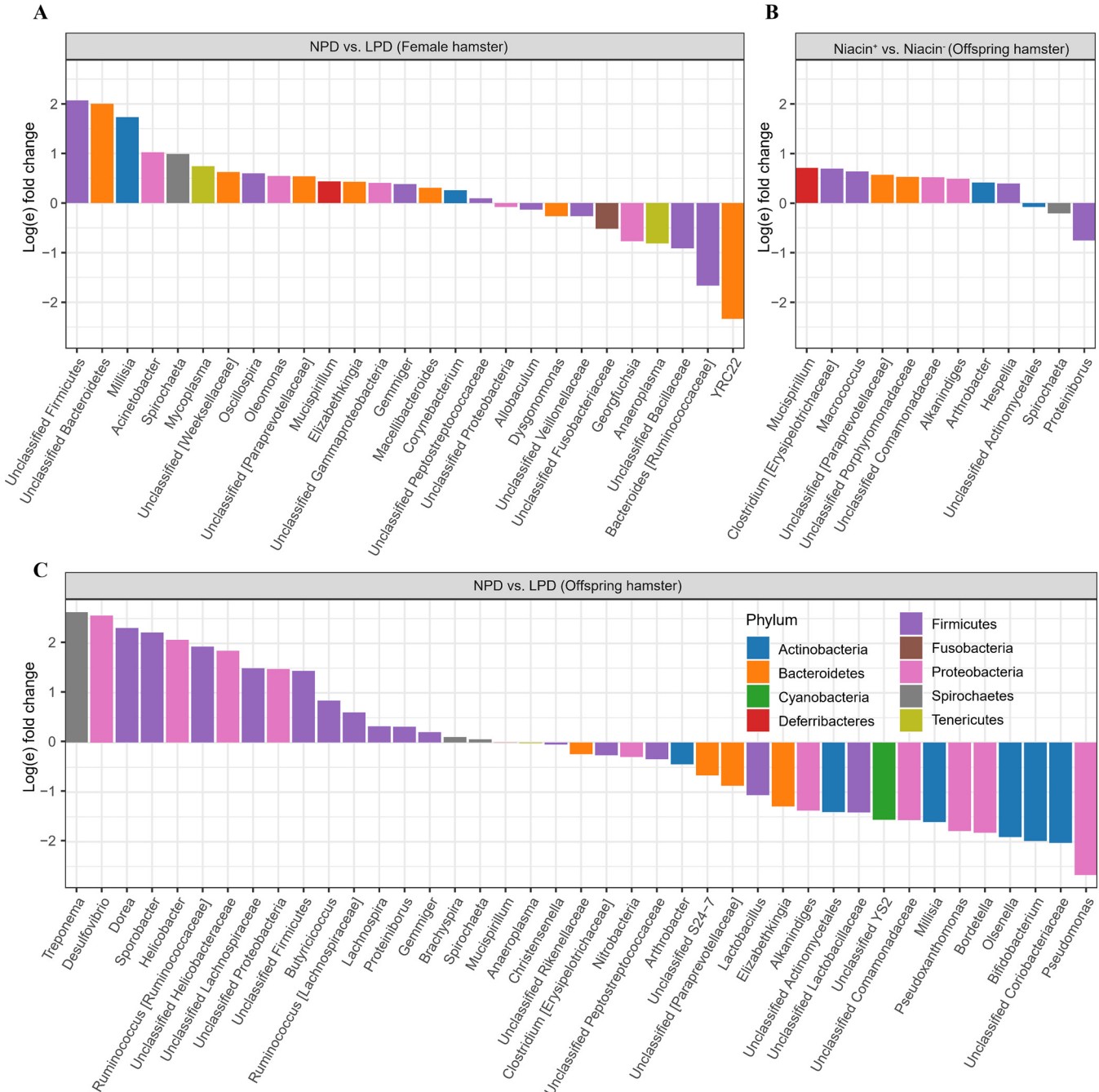

**FIG 5** Significant differential abundance of fecal microbial genera (log fold change). (A) Normal-protein diet (NPD) versus low-protein diet (LPD) in fecal microbiota of female hamster; (B) niacin-supplemented diet (Niacin⁺) versus niacin-deficient diet (Niacin⁻) in fecal microbiota of offspring hamster; (C) NPD versus LPD in fecal microbiota of offspring hamster.

those from studies of the European hamster (3). In this study, we found that a maternal LPD significantly and negatively affected mating behavior, reproduction, and body growth of female hamsters. A maternal LPD significantly increased the number of cohabitation tests for a female to successfully mate with a male and mount latency but decreased litter size (Fig. 2C, D, and E). We speculate that a maternal LPD may impair the development of the nervous system and then disrupt the cognitive ability of female hamsters, which is essential for mating. In our study, we did not find the significant effect of niacin deficiency on reproduction performance of female hamsters, which is different from findings in the European hamster (3). The difference may be

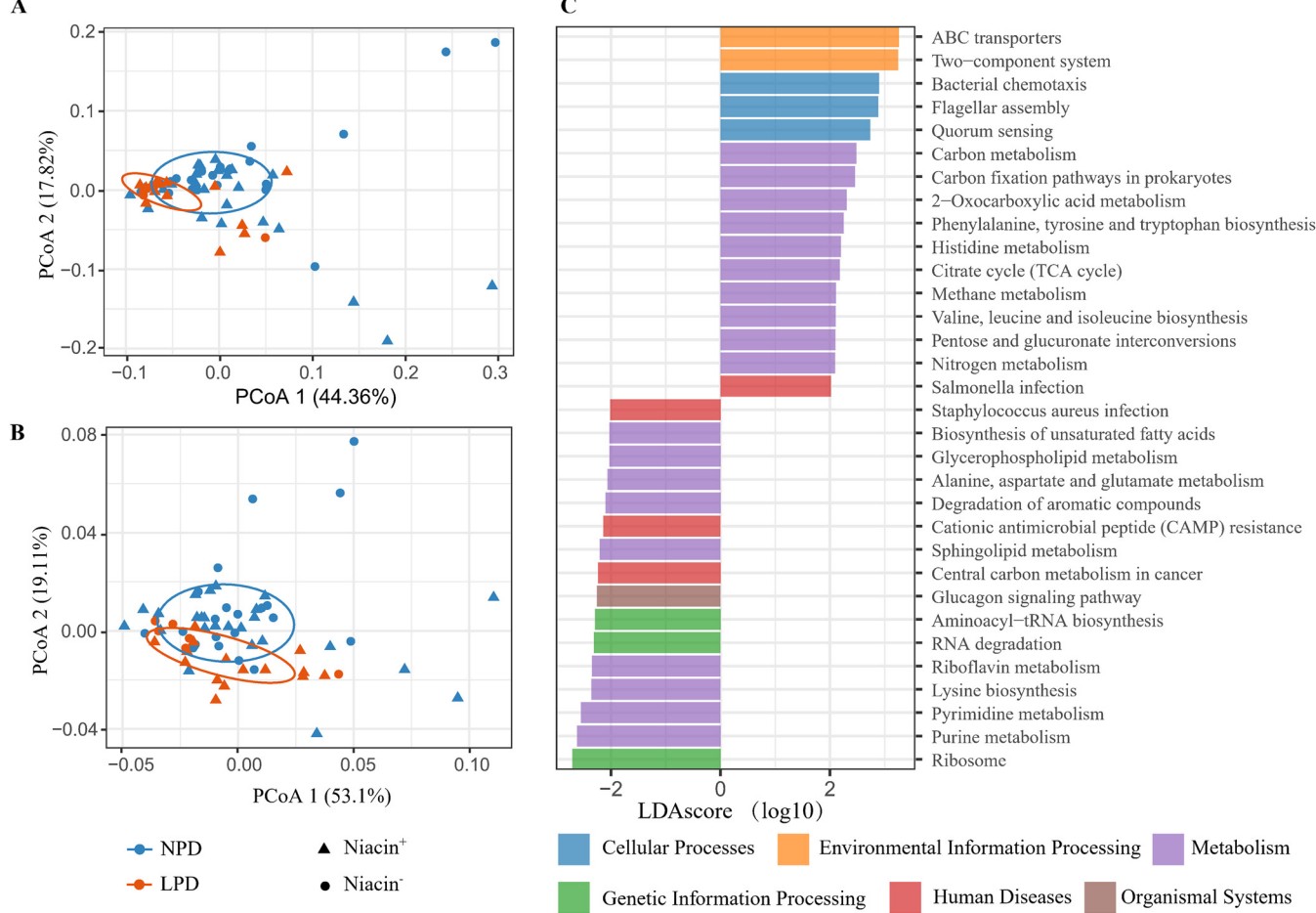

**FIG 6** Differences in functional orthologs and KEGG pathways of the fecal microbial community in offspring hamsters between maternal normal-protein diet (NPD) and low-protein diet (LPD) groups by using PCoA. (A) Predicted KO (KEGG Orthology) functional orthologs; (B) KEGG pathways. (C); differential KEGG pathways between NPD and LPD. Negative and positive LDA scores represent those pathways enriched in offspring hamsters of maternal LPD and NPD groups, respectively. Abbreviations: Niacin⁺, niacin-supplemented diet; Niacin⁻, niacin-deficient diet.

caused by species due to a difference in physiological demands for niacin or in the biosynthesis of niacin during the perinatal period (40).

**Effects of protein deficiency on the gut microbial community of hamsters.** An LPD could disrupt homeostasis of gut microbiota and result in a set of disorders (26, 28). A few previous studies have evaluated the response of the gut microbiome to dietary protein shift (29). However, results representing the effects of protein on the gut microbiome are not consistent, probably due to differences in protein concentration or protein source or species (28, 41–43). An LPD may increase $\alpha$ diversity of microbiota in rodents (41) but decrease $\alpha$ diversity in insects (42), while in a cohort study in humans, no significant effect of dietary protein on microbial $\alpha$ diversity was found (43). Nonetheless, lower diversity of gut microbes in humans is found to be linked to many diseases, such as obesity and inflammatory bowel disease (IBD) (19). In this study, we found maternal protein deficiency significantly affected the microbial community in both female hamsters and their offspring (Table 1 and Fig. 3), and $\alpha$ diversity of offspring was significantly reduced (Fig. 3C), which was associated with their metabolic function pathways (described below). Because gut microbes play an essential role in normal function and health of hosts (17–19), the LPD-altered microbes appear to have a negative effect on body growth in offspring, but this needs further investigation (44).

Based on results of predicted pathways (Fig. 6C and see Table S9 in the supplemental material), we observed a significant decrease in protein metabolism (phenylalanine, tyrosine, tryptophan, valine, leucine and isoleucine biosynthesis, and histidine metabolism),

protein machinery (ABC transporters), signal transduction (two-component system), and cell motility (bacterial chemotaxis and flagellar assembly) in maternal LPD offspring. These pathway depletions may be associated with the reduction of bacterial diversity in the offspring. However, no such difference was found in female hamsters, suggesting the effects of protein deficiency were more pronounced in the early development of offspring during pregnancy or the lactation period. The effects of LPD on microbial structure in female hamsters could be partially weakened by other factors, such as a compensatory decrease in muscle breakdown, increased food intake, or strengthened proteolytic activity in the large intestine (28, 45). Increasing evidence suggests that mother gut microbiota is important in fetus development and breastfeeding (31, 32, 46). The deterioration of milk quality in female hamsters may also directly influence gut microbiota in offspring, which may result in a decline of carbohydrate and amino acid metabolism processes as observed in this study (Fig. 6C and Table S9). Further studies based on manipulation of gut microbiota are needed to elucidate whether the alteration seen in the offspring's microbiota was caused by the mother hamsters' microbiota.

*Lactobacillus* and *Bifidobacterium* account for ∼17.20% and 0.60% of all microbes found in the total fecal samples in our study. Many species in these genera were recognized as probiotics to prevent or treat gastrointestinal diseases or alleviate behavioral disorders (47, 48). Breastfeeding is the key postnatal link between mothers and neonates and drives the microbial colonization (49, 50). A previous study reported that an LPD in mice after weaning could reduce the mucosal colonization of *Lactobacillus* and inhibit its recognition by IgA (51). We found maternal LPD significantly increased abundance in *Lactobacillus* and *Bifidobacterium* in the offspring (Fig. 5C and Table S6). The difference may be caused by difference in milk intake of the offspring during the lactation period, which needs further investigation.

We found abundance of the top six taxa, including *Treponema, Desulfovibrio, Sporobacter, Helicobacter, Dorea*, and *Ruminococcus*, was decreased but probiotics were enriched in offspring of maternal LPD group (Fig. 5C and Table S6). No such effects were observed in offspring of maternal niacin$^-$ diet group. These observations suggested that gut microbiota of hamsters was more sensitive to dietary protein deficiency than to niacin deficiency. Lower abundance of *Desulfovibrio* was correlated with lower body mass index (BMI) (52) and birth weight (53) in humans, which is consistent with our observation that lower body weight of offspring was also associated with lower abundance of *Desulfovibrio*. Moreover, we observed that genera with a larger change in abundance like *Unclassified Lachnospiraceae, Dorea*, and *Ruminococcus* were fermented bacteria (Fig. 5C and Table S6), which contain various short-chain fatty acid (SCFA)-producing species (54–58). SCFAs are primary end products fermented from dietary fibers by gut microbiota, which play an important role in modulating gut homeostasis, immune function, and even gut-brain communication (56, 58). Disturbance of SCFA producers could be caused by the decreased nutritional content in offspring, which may detrimentally impact the colon health of animals (47, 57). Therefore, reduction of these important SCFA-producing microbes in the LPD group may impose negative effects on both female hamsters and their offspring.

**Effects of niacin deficiency on gut microbiota of hamsters.** Niacin (also called vitamin B$_3$), can be found in several forms, like nicotinamide and nicotinic acid, and is required in multilevel cellular processes. Mammals are able to absorb niacin from food or biosynthesize it from tryptophan (5, 40). Niacin can exert beneficial effects on maintaining gut health (11, 30). However, the effects of niacin on an offspring's gut microbiota have rarely been investigated. In this study, we found that maternal niacin-deficient diets significantly altered the gut microbial community in offspring hamsters (but not in adult female hamsters). It is likely that niacin is synthesized from other substrates, such as tryptophan (maize starch and casein contains low levels of tryptophan), in the diet of maternal animals; therefore, the reproduction and gut microbiota are less affected by niacin deficiency (40). Besides, maternal niacin deficiency may impact the gut microbial community through reduction in milk yield as demonstrated in another study (10). We observed that niacin deficiency significantly increased $\alpha$ diversity indices (Chao1 and observed features) of

offspring hamsters (Fig. 3D), which is contradictory to a previous study (59). We speculate that, in our study, maternal niacin deficiency may have stimulated the biosynthesis of niacin, which is involved in multiple bacterium species.

**Limitations of this study.** This study suffers several limitations that restrict the interpretation of some results. First, although nutrition deficiency may reduce the fitness of hamsters under laboratory conditions, the causal links between nutrition deficiency and sustained population decline of greater long-tailed hamsters under field conditions are still unclear. Thus, we are not sure if the observed negative effects of malnutrition on hamsters in the laboratory are applicable to field conditions. Future studies should examine the relationship between nutritional deficiency of hamsters in croplands due to intensified monoculture and their reproduction performance. Second, the roles of milk production and quality in offspring hamsters are still unknown, which would obscure the effects of malnutrition and gut microbes on offspring of hamsters. Therefore, their distinct impacts need further investigation. Third, the transfer of gut microbes between mother hamsters and their offspring was not assessed, which prevents us from evaluating the effects of maternal gut microbiota on offspring microbiota. Besides, the impacts of caging the animal or rodent chow on microbial input to offspring also need to be assessed. Finally, microbiota transplantation experiments and multiomic analysis are necessary to identify malnutrition-induced gut microbiome disturbances and to examine their potential impacts on physiological and reproductive performance of hamsters.

**Conclusion.** Our results demonstrate that maternal protein deficiency has a more profound negative effect on fitness and the gut microbes of hamsters than maternal niacin deficiency, and such an effect is more pronounced in offspring than in female hamsters. Our study suggests that malnutrition and associated gut microbes may be a potential factor in causing the population decline of the greater long-tailed hamster in the farmland of the North China Plain due to intensified monoculture. Conservation practices should therefore consider surveying the dietary species richness and host-associated microbiota of animals (60), which may help explain the bottom-up effects of human disturbance on their populations. Future studies should emphasize ways to overcome the above limitations of the study, so as to reveal the roles of gut microbes in affecting the fitness of greater long-tailed hamsters due to nutritional deficiency caused by intensified monoculture in farmlands.

## MATERIALS AND METHODS

**Experimental design.** The greater long-tailed hamsters (*T. triton*) used in this study were obtained from a laboratory breeding colony at the Institute of Zoology, Chinese Academy of Sciences (CAS). A total of 102 sexually naive hamsters (51 of each sex, 6 to 7 months of age) from different litters were used in this study. *T. triton* is a solitary, polygamous rodent species with intense aggressive behavior during conspecific encounters. Individuals were separately housed in polypropylene cages that contained corncob fragments as bedding for 1 month ahead of the experiment and had free access to food (AIN93G rodent diet, Beijing KeAo Bioscience Co.) and water. All cages were maintained under a 16-h light/8-h dark cycle, and the temperature in the animal room was maintained at 22 $\pm$ 2°C. We designed a 2- by 2-factor experiment to test the effects of maternal protein and niacin deficiency on female hamsters and their offspring (Table 2). The diet formula was modified based on standard AIN93G rodent chow. Female (maternal) hamsters were randomly allocated to four diet treatment groups with combination of low-protein diet (LPD) or normal-protein diet (NPD) and niacin supplement (niacin$^+$) or niacin deficiency (niacin$^-$): (i) normal-protein and niacin-supplemented group, (ii) low-protein and niacin-supplemented group, (iii) normal-protein and niacin-deficient group, and (iv) low-protein and niacin-deficient group (for sample size, see Table 2). Adult males used for mating with maternal females were fed with standard AIN93G rodent chow during the study (Fig. 1C).

Adult female hamsters (*n* = 51) were fed with the above modified rodent chow for 1 month before the breeding experiment (Fig. 1). Body weight of the adult female hamsters was measured at days 0, 6, 10, 14, 18, 22, and 30 to represent the body growth condition of female hamsters. As *T. triton* displays very intense aggressive behavior toward conspecifics or people, to minimize handling stress for hamsters, the length of body or tail was not measured as the body growth indicator in this study. After a 30-day experimental period, fecal samples of female hamsters were collected for 2 days to assess the effects of different diets on the maternal female's gut microbiota (Fig. 1A). We used an additional neutral arena box, following a previous study (61), to conduct the cohabitation tests to measure the reproduction performance of female hamsters (Fig. 1B). The box was divided into two cells with a movable partition (see the black plate in Fig. 1B); each cell contains a fixed protective screen (see the small gray plate in Fig. 1B) as a shelter to avoid intense biting between hamsters. The cohabitation box was cleaned using water

and 75% ethanol between cohabitation tests to eliminate odor and residue. The experiment was conducted in the first 4 h at the beginning of the dark cycle. An adult female (with an opened, moist, and pinkish vagina) and a randomly selected male were placed in the two cells of the box (80 by 80 by 100 cm) for a 5-min acclimation period, and then the middle partition of the box was removed to start the cohabitation test. Once the hamster pair in the cohabitation box displayed mating behavior (mounting, thrusting, and intromission) more than three times in 15 min, the female was no longer assigned to mate with other males and housed individually to facilitate pregnancy and giving birth. If the female was not able to successfully mate with the male, then another male was introduced into the cohabitated box the next day or when another estrous cycle for the female began. If no mating behavior of the hamster pair was observed in 14 days, or the female was injured during the cohabitation test, she was excluded from further analysis. In total, nine females died or were injured during the study and 19 females gave birth (although one case of infanticide was observed under the NPD-niacin⁻ group [see Table S1 in the supplemental material]). The pregnancy period of a hamster lasted 26 to 28 days, with litter sizes ranging from 1 to 6. The lactation period of litters lasted 26 to 27 days. Body weights of the total 68 pups from four treatment groups (5, 5, 5, and 3 litters from normal-protein and niacin⁺, normal-protein and niacin⁻, low-protein and niacin⁺, and low-protein and niacin⁻ groups, respectively) were sampled to represent the body weights of offspring hamsters.

Mating behaviors, including mount latency, mount frequency, number of cohabitation tests, and litter size were recorded to represent the reproduction performance of female hamsters. Mount latency was defined as the time needed for a female to have the first copulation with a male that displayed mounting, thrusting, and intromission while the female showed a typical lordosis posture. Mount frequency was defined as the number of copulations of the hamster pair within 15 min. The cohabitation test ended when the chasing behavior of male individuals was not observed or the female started to attack the male after several copulation attempts. Litter size was defined as the number of pups of a female hamster at birth. The number of cohabitation tests was defined as the number of days required for a female to successfully mate with a male. Higher mount latency and number of cohabitation tests, lower mount frequency, and smaller litter size indicate a reduction of reproduction performance in a female hamster.

**Fecal sample collection.** Fecal samples from the female hamsters were collected within 2 days before cohabitation tests, while offsprings' fecal samples were collected immediately (within 1 to 2 days) when they were separated from the female hamsters at weaning (Fig. 1A). Hamsters were moved to a clean empty cage with a bottom of stainless-steel mesh in order to collect fecal samples. Fecal boli were collected immediately and stored at −80°C for DNA extraction. In total, 36 fecal samples from 45 female hamsters and 68 fecal samples from 68 offspring hamsters at weaning were collected, stored, and used for further analysis. One litter in the niacin deficiency group was removed from amplicon analysis because of maternal infanticide (Table S1).

The animal raising and handling were in line with guidance by the Animal Care and Use Committee of Institute of Zoology, Chinese Academy of Sciences. The authors that conducted the animal experiments were trained by the Beijing Agency for Experimental Animals, China, with an authorized diploma.

**Amplicon sequencing and analysis.** The total DNA of the fecal samples was extracted using a NucleoSpin 96 Soil kit (Macherey-Nagel, Germany) based on the manufacturer's protocol. DNA concentration was measured by fluorometry using the Qubit double-stranded DNA (dsDNA) assay kit and fluorometer (Life Technologies, Carlsbad, CA, USA). The V3-V4 region of the 16S rRNA gene was amplified for 20 cycles using universal primers (forward primer 338F, 5′-ACTCCTACGGGAGGCAGCA-3′; reverse primer 806R, 5′-GGACT ACHVGGGTWTCTAAT-3′) with the primers containing adapter and barcode sequences. PCR amplification was performed with the following thermocycling conditions: an initial denaturation at 95°C for 5 min, followed by 15 cycles at 95°C for 1 min, 50°C for 1 min, and 72°C for 1 min, with a final extension at 72°C for 7 min. The PCR products from the first PCR step were purified using VAHTS DNA Clean Beads, and a second round of PCR was performed in a 40-$\mu$L reaction mixture containing 20 $\mu$L 2× Ph$\mu$sion HS (high-fidelity) master mix, 8 $\mu$L double-distilled water (ddH$_2$O), 10 $\mu$M each primer, and 10 $\mu$L of PCR products from the first step. The thermocycling conditions for the second round of PCR were as follows: an initial denaturation at 98°C for 30 s, followed by 10 cycles at 98°C for 10 s, 65°C for 30s, and 72°C for 30 s, with a final extension at 72°C for 5 min. Finally, all PCR products were quantified using Quant-iT double-stranded DNA (dsDNA) HS (high-sensitivity) reagent and were pooled. High-throughput sequencing of bacterial rRNA genes was performed with the purified, pooled sample using an Illumina HiSeq 2500 platform (2 × 250 paired ends) at Biomarker Technologies Corporation, Beijing, China. Sterile water was used as the negative control in DNA extraction and PCR amplification.

Bioinformatic analysis was conducted with QIIME2 (Quantitative Insight into Microbial Ecology, version 2022-2) (62) software. Raw data FASTQ files were transformed into a format that could be read by the QIIME2 system using the *qiime tools import* program. Demultiplexed sequences from each sample were quality filtered and trimmed (left at 19 bp and right at 20 bp based on primer length), denoised, and merged using the *DADA2* plugin (63). Any contaminating mitochondrial and chloroplast sequences were filtered using the QIIME *feature-table* plugin. The feature table of amplicon sequence variants (ASVs) was obtained by *qiime tool export* function with table.qza generated by the *DADA2* plugin. Representative sequences were obtained by the *feature-table* plugin. A total of 5,792 ASVs were kept for further analysis. Taxonomy was assigned using the naive Bayes feature classifier trained against the Greengenes 13_8 database (64). Diversity metrics were calculated using the *qiime diversity alpha* plugin without rarefaction (for feature counts, an average of 22,220, minimum of 9,963, and maximum of 45,253); sequence depth was included as a model covariate in the analysis. $\alpha$ diversity indices, including Shannon diversity, observed features (the number of observed features for each sample), Chao1 richness estimator, and Faith's phylogenetic diversity index were calculated to estimate the microbial diversity within an individual sample. Unless specified above, the parameters used in the analysis were set as default.

**Statistical analysis. (i) Analysis of body weight and reproductive behaviors.** Effects of protein and niacin diets in the 2- by 2-factor design on body weight change of female hamster were analyzed

by linear mixed model (LMM) in R (65). We used body weight as the response variable, maternal diet treatment and sampling time as the fixed effects, and hamster identity as the random effect. Differences in body weights of female hamsters between diet groups (different protein and niacin levels) at the same sampling time were tested using *aov* (analysis of variance). We also performed repeatability estimation of body weight data using a mixed-effects model by the *rpt* function in package *rptR* (66, 67); the estimate of weight measurement was shown to be highly repeatable (repeatability = 0.869; 95% confidence interval [CI], 0.804, 0.908). Normality of body weight was tested using the Shapiro-Wilk test ($P = 0.02$). Differences in offsprings' body weights between diet groups were tested using the *aov* function after normality test (Shapiro-Wilk test, $P > 0.05$).

Effects of protein and niacin diets and their interaction on mount latency, mount frequency, and number of cohabitation tests between diet groups were analyzed using the generalized linear model (GLM) with Poisson distribution probability (link function, log). GLM with binomial distribution (link function, logit) was performed to test the difference of sex ratio of offspring between different diet groups. The *Anova* function in the package *car* was used to test the significance of variables in GLMs. $R^2$ values of GLMs were calculated with the *rsq* package in R.

**(ii) Analysis of $\alpha$ and $\beta$ diversities.** Differences in $\alpha$ diversity, including Shannon diversity, Chao1 index, observed features (estimates of total features), and phylogenetic diversity between different diet groups, were tested using a two-way analysis of covariance (ANCOVA) after a normality test (Shapiro-Wilk test, $P > 0.05$) in R; sampling depth was included as a covariate to control for differences in library size.

Analysis of compositional data referred to a guide described by Quinn et al. (68). A Bayesian multiplicative replacement strategy was conducted to handle the zero counts in the feature table by package *zCompositions* (1.4.0-1) (69) in R, and then the feature tables were subjected to clr (central log ratio)-transformation by the *composition* (v.2.0-4) package. Euclidean distance matrices were computed from transformed ASV table by the *vegan* package. Analysis of multivariate homogeneity of group dispersions was conducted in the *vegan* package (70) in R, while permutational multivariate analysis of variance (PERMANOVA) of the mixed-effects model based on Euclidean distance matrices was performed with PRIMER 7 software (71, 72) (with 999 permutations of residuals under a reduced model and a type I sum of squares). Protein and niacin levels were treated as fixed factors, while the dam's ID was treated as random factor in the model. Principle-coordinate analyses (PCoAs) were also performed to visualize the dissimilarity of gut microbial composition between samples based on Euclidean distance matrices.

**(iii) Analysis of differential abundance at genus level.** The ANCOMBC (analysis of compositions of microbiomes with bias correction) algorithm controlled the false-discovery rate (FDR) better than other methods (73). Differential abundance analysis of genus taxa was performed using ANCOM-BC methodology in R package *ANCOMBC* (73). The taxonomy table, feature table, phylogenetic tree file, and metadata table were merged into a phyloseq object by the *phyloseq* package (74) and then used for the ANCOMBC algorithm. Taxa with proportions of zeroes greater than 0.9 in all samples were dropped from the analysis. $P$ values were adjusted with the Holm procedure, and the cutoff threshold of the adjusted $P$ value was 0.05.

**(iv) Microbial functional profiling.** To further investigate the effects of protein and niacin deficiency on functional repertoires of gut microbes, we used the ASV unique sequences and feature table to infer function orthologs and KEGG pathway profiles. The relative abundance of predicted functions and pathway of each sample were gained using the *Tax4fun2* package (75) with default settings in R. The *Betadisper* function in the *vegan* package was used to analyze multivariate homogeneity of group dispersions. Differences in profiles of KO functional orthologs and KEGG pathways were computed with PRIMER7 software. PCoA was performed here to depict sample dispersions and variations between dietary treatment groups. PERMANOVA and PCoA were both conducted using the Bray-Curtis dissimilarity matrix. Downstream differential analysis of KEGG functions and pathways was conducted in LEfSe (76) with default parameters. (The threshold for the LDA score was 2.0, and $P$ values were adjusted with the Bonferroni procedure.)

**Data availability.** Raw sequence data can be acquired from the NCBI Sequence Read Archive (SRA) under BioProject accession no. PRJNA816668.

## SUPPLEMENTAL MATERIAL

Supplemental material is available online only.
**SUPPLEMENTAL FILE 1**, PDF file, 0.4 MB.

## ACKNOWLEDGMENTS

This study was supported by the Strategic Priority Research Program of Chinese Academy of Sciences (grant no. XDPB16, XDB11050300), the International Partnership Program of Chinese Academy of Sciences (grant no. 152111KYSB20160089), and the External Cooperation Program of BIC, Chinese Academy of Sciences (grant no. 152111KYSB20150023).

We declare no conflict of interest.

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
