## [Reviewer comments · Microbiology Spectrum]

Microbiology Spectrum

Impacts of dietary protein and niacin deficiency on reproduction performance, body growth and gut microbiota of female hamsters (*Tscherskia Triton*) and their offspring

Jidong Zhao, Wei Lu, Shuli Huang, Yvon Le Maho, Caroline Habold, and Zhibin Zhang

Corresponding Author(s): Zhibin Zhang, Institute of Zoology, Chinese Academy of Sciences

Review Timeline:

Submission Date:	January 16, 2022
Editorial Decision:	March 7, 2022
Revision Received:	June 3, 2022
Editorial Decision:	August 7, 2022
Revision Received:	September 28, 2022
Accepted:	October 10, 2022

Editor: Kevin Theis

Reviewer(s): Disclosure of reviewer identity is with reference to reviewer comments included in decision letter(s). The following individuals involved in review of your submission have agreed to reveal their identity: Karina Montero (Reviewer #1); Dominik Schmid (Reviewer #2)

Transaction Report:

DOI: <https://doi.org/10.1128/spectrum.00157-22>

March 7, 2022

Dr. Zhibin Zhang
Institute of Zoology, Chinese Academy of Sciences
Beijing
China

Re: Spectrum00157-22 (Maternal low protein and niacin deficiency diet induce growth retardation and dysfunction of gut microbiota in the offspring of a rodent)

Dear Dr. Zhibin Zhang:

Thank you for submitting your manuscript to Microbiology Spectrum. The manuscript has been reviewed by two experts in the field. Based on their comments, which are included below, major revisions would be required for the manuscript to be further considered for publication.

If you do choose to address the reviewer comments and resubmit the manuscript, please provide (1) point-by-point responses to the issues raised by the reviewers as file type "Response to Reviewers," not in your cover letter, and (2) a PDF file that indicates the changes from the original submission (by highlighting or underlining the changes) as file type "Marked Up Manuscript - For Review Only". Please use this link to submit your revised manuscript - we strongly recommend that you submit your paper within the next 60 days or reach out to me. Detailed instructions on submitting your revised paper are below.

Link Not Available

Sincerely,

Kevin R. Theis

Journals Department
Reviewer comments:

Reviewer #1 (Public repository details (Required)):

16S rRNA gene raw sequences need to be made available in the NCBI Sequence Read Archive (SRA)

Reviewer #1 (Comments for the Author):

Zhao et al. use an experimental approach to test whether reduced levels of dietary components (protein and niacin) influence

aspects of the reproductive biology and diversity of the gut microbiome of greater long-tailed hamsters.

I think that some sections of the manuscript would benefit from re-structuring to improve clarity. This is particularly true for the Introduction. There are a number of issues concerning both bioinformatics and statistical analyses (eg. the use of cluster instead of error-model -based methods to process sequence variants). In addition, there is a lack of detail regarding data collection. Altogether, these issues make me question the robustness of the results and I suggest that the authors revise the manuscript to improve transparency and clarity. Finally, there are a number of confounding factors that could influence the outcomes of this study. Some of these can be controlled for statistically (i.e. include sex and litter size as explanatory variables in the relevant analyses). But some confounding effects intrinsic to the experimental approach (i.e. legacy effects of housing) are not addressed. The authors should discuss the limitations of their study. I develop these and other concerns and suggestions on how to improve the manuscript in detail below. I hope the authors find these comments useful.

INTRODUCTION

A main premise of the study is that nutritional deficiency has a negative effect on both reproduction and gut microbiome diversity of greater long-tailed hamsters and that such findings support the notion that nutritional deficiency is a major driver of population decline of this species. Indeed, a large section of the introduction focuses on this issue. Whilst I agree that this is a very relevant question, the approach used in this study (e.g. the use of lab-reared animals, punctual sampling of fecal samples, etc.) falls short in its scope to answer it. Long-term monitoring of natural populations or a population level study comparing sites with a gradient of nutritional stressors, for instance, seems more appropriate to evaluate the role of agricultural practices on fitness and whether they contribute to population declines. It seems like most of the introduction is off-topic. Therefore, I would suggest that the authors instead place greater emphasis on developing specific hypothesis regarding experimental manipulation of nutritional factors and their effect on reproduction and host-associated microbiomes.

RESULTS

L. 111-112: In the context of fitness estimates, body condition represents a more biologically informative measure than weight (see e.g. Peig & Green 2009 *Oikos*). Also, please report estimates of repeatability.

Figure 1B: this figure is not clear, what does "reproductive output" represent? Why use proportions?

L. 129: Do "CD" stand for Control treatment? Should it be CON? If so, please be consistent throughout the text.

L. 129-133: Please use a table to report the group-specific (i.e. treatment) estimates (including R²) based on PERMANOVA analysis, not sure what is being reported here. Also, based on the results, I do not think you can argue that groups are "strongly separated" (especially since R² estimates are not reported). Indeed, Figure 2 A does not suggest differentiation at all...

PCoA s (Figure 2 and 5): there seem to be very large differences in beta-dispersion between groups (particularly among offspring samples) - this violates a key assumption of PERMANOVA that groups have similar dispersions. The authors should report if there are significant differences in beta-dispersion between groups.

L. 134-141: Did you control for differences between female and male offspring? The litter size?

L.164-172: As with taxonomic based analysis, please follow the recommendations above for reporting the results of multivariate analysis on functional predictions..

MATERIALS AND METHODS

Experimental design:

How many different genotypes were used in this study? Do all females come from different litters or are some individuals closely related?

It is not clear what are the final sample sizes across treatments and how many individuals were used for each analysis. For example, L. 341 "44 females were separately housed..", L. 345 "51 female hamsters were randomly allocated to four diet groups", while Table 1 summarizes results from a subset of 46 individuals. This is quite confusing. A table would make this information more accessible and transparent.

Sample collection:

Were all fecal samples from females collected at the same time? How old were they?

Offspring fecal samples were collected after weaning, but its not clear how old was each individual sampled. Was the timing of weaning different across treatments? Please provide more details.

How many individuals from the "offspring" group were female / male? How many individuals were sampled per litter?

Amplicon sequencing:

What is the sequencing depth (total and per sample average/range)?

How many OTUs were kept after prevalence and abundance thresholds were applied to the data?

More importantly, were negative controls (extraction and PCR) used?

Clustering sequences based on a threshold (OTUs) provide limited resolution. Moreover, clustering (OTU) vs error-model based (ASV) have been shown to influence biological interpretations (e.g. Joos et al. 2020, BMC Genomics; Chiarello et al. 2022 PlosOne). I would suggest that authors use a more up-to date approach for down-streaming analysis (i.e. dada2).

Also, please make the 16S rRNA gene raw sequences available in the NCBI Sequence Read Archive (SRA)

Statistical analysis:

L. 395 - 399: Please use generalized linear models (with non-normal errors) instead of Kruskal-Wallis test or a Fisher exact test. In addition, please report estimates of effect size.

L. 401-403: PcoA is a multivariate method used for exploration and visualization, but it does not allow inference of statistical significance- for this PERMANOVA is commonly used.

Also, not sure what "the significance of the beta diversity" mean. I would suggest rewording, something in the lines of "Test for differences in community structure according to nutritional treatment".

L. 405-418: Was rarefaction applied? How do authors control for sample read depth? This information is crucial, especially when using a method like LefSe.

A major challenge of amplicon-based 16S profiling of bacterial communities is that relative abundance corresponds to compositional data with much debate surrounding which method is more appropriate when evaluating "differential abundance" (also, see Nearing et al. 2022 NatComm for further discussion and implications around this topic). Please use a method that properly address this issue (e.g. Morton et al. 2019 NatComm; Mandal et. al. 2015 MicrEcol.; Lin & Das Pedada 2020 NatComm).

DISCUSSION

I think the authors need to be more careful with regard to inferring causality. For instance in L192-196 "cause disorder of their gut microbiota" and L 194-195 " amplified in offspring via dysfunction of microbiota". With this data, it is not possible to infer dysbiosis (see eg. Hooks & O'Malley 2017 mBio). In this context, I would also urge the authors to soften the title of the manuscript and to not infer causality.

Also, there is no discussion of the limitations of this study. A major concern is that even if environmental factors are kept constant in the lab, variation in microbial input (i.e. cage or legacy effects) is likely to be an important (if not the main) factor influencing the findings reported in this study. This should be acknowledged and properly discussed.

There were a number of grammatical errors throughout the text, including missing punctuation and awkward wording.

These are some examples but there are many more in the text:

L. 56: Should be "rodent species are vital" instead of "rodent species is vital"

L. 58: Should be "indiscriminate" instead of " indiscriminative".

L. 60: Should be "facing sharp declines and extirpation" instead of "facing sharp decline or even extinct at"

L. 341, L. 345: Avoid starting a sentence with a number.

L. 362: Should be "In total," instead of "Totally,"

L. 363: Should be "due to death" instead of "due to dead".

L. 370: " fecal samples" instead of " fecal sample."

Reviewer #2 (Comments for the Author):

The study by Zhao submitted for publication in Microbiology Spectrum aims to understand the transgenerational fitness effects of nutritionally poor diets. The experiment (which isn't clear early on in the manuscript) contrasts mating behaviour, maternal and offspring gut microbial diversity in groups of hamsters fed with either protein or niacin deficient diets. The authors loosely tie this to threats to rodents in anthropogenically altered landscapes. While I think the study has intrinsic merit and tackles an interesting idea, the ideas, reasoning and presentation of their results are confusing and hard to follow. The experimental design and statistical evaluation are also unclear. As such, I cannot recommend this study to be published without major revisions.

Check grammar, phrasing and spelling throughout. It is worth considering proof-reading by an independent colleague or service.

Here are some more specific comments

Abstract:

L16: The first two sentences do not link well with one another and do not spell out strongly enough the logical connection between maternal effects, nutrient deficiency and intensified agricultural practices.

L24: Replace "Greater long-tailed hamsters" with either "hamsters" or T. triton.

L24 and following: There was no background or reasoning as to why microbiota was looked at nor explained when you discuss the study question and design. How did you quantify less responsive?

L30: grammar issues. Its "increases" and "decreases"

L33: nowhere was explained that this study is an experiment rather than a field study. Those details need to be given in the abstract. The links between protein deficient diet, niacin deficiency and monoculture are still obscure even at the end of the abstract.

Importance:

L37: the first two sentences here are a much better line of argument than in the abstract.

L38: I think this sentence is poorly incorporated. There is a logical leap from population declines to maternal effects (which are a very small factor overall contributing to offspring fitness)

L42. Grammar issues in this sentence

Introduction

L50: "making some of them 53 successful living organisms in the earth" - poor word choices and phrasing.

L56: "is" = "are"

L58: what do you mean with proper? This sentence does make very little sense

L72: the paragraph ending here should be at least two:

- 1) agricultural intensification and its impact on biodiversity
- 2) rodent diversity and its functional importance + threats from monoculture

L73: poor phrasing; the paragraph feels confused as to what it wants to say; why mention disease here if your main focus is diet? If you want to talk about disease, you can do so in the discussion and suggest that poor fitness and gut microbial diversity can also lead to more severe diseases/infections

L87: this is very confusing. Unfortunately, you have not built a case as to how "maternal dysfunction of gut microbiota" could even affect offspring development.

L102: Don't use apparently.

Results:

L111: on which response variable. Since you are submitting to a journal with a different, short format that puts the methods at the end, you still need to write the results with as much information as needed to understand what you found.

L115: its still unclear to me what you done. The last paragraph of the introduction needs to also specify what type of experiment was done etc - there is so little information to go by. Only by looking at the methods section, I can see what you did, but this does not work in a format like this.

L118: indicate the direction of your effect, i.e., increase/decrease etc

L122: Is this behaviour in one group meaningful, i.e., did you test for its effect?

Overall, the graphs look nice. Maybe consider polygons for the different diet groups in the PCoA plot.

I have to reiterate that the link between maternal fecal microbiome and offspring is not clear: to spell it out, for me, the link is, that differently fed mothers inherit, i.e., seed their offspring with different microbes. But what is not clear to me is whether the offspring are feeding themselves on maternal milk or also the diet. Can't even get that from the methods.

L164: if neither of the treatments impacted microbial functions, how do you then argue that the treatments have an effect on host fitness?

L172: It concerns me that with two comparable functions you end up with two contrasting results.

Discussion

This is likely to change given results and method changes so I wont comment on this now.

Methods

There is two little information on the microbiome sequencing and data processing (e.g.,

reads, filtering parameters

Is this QIIME or QIIME2 - there are differences between the versions and I would recommend the newer one.

Why where some stat test calculated in QIIME and others in R. The reporting of the statistical test is hard to follow.

L390: grammar

Staff Comments:

Preparing Revision Guidelines

Please return the manuscript within 60 days; if you cannot complete the modification within this time period, please contact me. If you do not wish to modify the manuscript and prefer to submit it to another journal, please notify me of your decision immediately so that the manuscript may be formally withdrawn from consideration by Microbiology Spectrum.

Review: Maternal low protein and niacin deficiency diet induce growth retardation and dysfunction of gut microbiota in the offspring of a rodent

The study by Zhao submitted for publication in Microbiology Spectrum aims to understand the transgenerational fitness effects of nutritionally poor diets. The experiment (which isn't clear for the longest time reading this study) contrasts mating behaviour, maternal and offspring gut microbial diversity in groups of hamsters fed with either protein or niacin deficient diets. The authors loosely tie this to threats to rodents in anthropogenically altered landscapes. While I think the study has intrinsic merit and tackles an interesting idea, the ideas, reasoning and presentation of their results are confusing and hard to follow. The experimental design and statistical evaluation are also unclear. As such, I cannot recommend this study to be published without major revisions.

Check grammar, phrasing and spelling throughout. It is worth considering proof-reading by a native.

Here are some more specific comments

Abstract:

L16: The first two sentences do not link well with one another and do not spell out strongly enough the logical connection between maternal effects, nutrient deficiency and intensified agricultural practices.

L24: Replace "Greater long-tailed hamsters" with either "hamsters" or *T. triton*.

L24 and following: There was no background or reasoning as to why microbiota was looked at nor explained when you discuss the study question and design. How did you quantify less responsive?

L30: grammar issues. Its "increases" and "decreases"

L33: nowhere was explained that this study is an experiment rather than a field study. Those details need to be given in the abstract. The links between protein deficient diet, niacin deficiency and monoculture are still obscure even at the end of the abstract.

Importance:

L37: the first two sentences here are a much better line of argument than in the abstract.

L38: I think this sentence is poorly incorporated. There is a logical leap from population declines to maternal effects (which are a very small factor overall contributing to offspring fitness)

L42. Grammar issues in this sentence

Introduction

L50: "making some of them 53 successful living organisms in the earth" - poor word choices and phrasing.

L56: "is" = "are"

L58: what do you mean with proper? This sentence does make very little sense

L72: the paragraph ending here should be at least two:

- 1) agricultural intensification and its impact on biodiversity
- 2) rodent diversity and its functional importance + threats from monoculture

L73: poor phrasing; the paragraph feels confused as to what it wants to say; why mention disease here if your main focus is diet? If you want to talk about disease, you can do so in the discussion and suggest that poor fitness and gut microbial diversity can also lead to more severe diseases/infections

L87: this is very confusing. Unfortunately, you have not built a case as to how “maternal dysfunction of gut microbiota” could even affect offspring development.

L102: Don't use apparently.

Results:

L111: on which response variable. Since you are submitting to a journal with a different, short format that puts the methods at the end, you still need to write the results with as much information as needed to understand what you found.

L115: its still unclear to me what you done. The last paragraph of the introduction needs to also specify what type of experiment was done etc – there is so little information to go by. Only by looking at the methods section, I can see what you did, but this does not work in a format like this.

L118: indicate the direction of your effect, i.e., increase/decrease etc

L122: Is this behaviour in one group meaningful, i.e., did you test for its effect?

Overall, the graphs look nice. Maybe consider polygons for the different diet groups in the PCoA plot.

I have to reiterate that the link between maternal fecal microbiome and offspring is not clear: to spell it out, for me, the link is, that differently fed mothers inherit, i.e., seed their offspring with different microbes. But what is not clear to me is whether the offspring are feeding themselves on maternal milk or also the diet. Can't even get that from the methods.

L164: if neither of the treatments impacted microbial functions, how do you then argue that the treatments have an effect on host fitness?

L172: It concerns me that with two comparable functions you end up with two contrasting results.

Discussion

This is likely to change given results and method changes so I wont comment on this now.

Methods

There is two little information on the microbiome sequencing and data processing (e.g., reads, filtering parameters

Is this QIIME or QIIME2 – there are differences between the versions and I would recommend the newer one.

Why where some stat test calculated in QIIME and others in R. The reporting of the statistical test is hard to follow.

L390: grammar

Microbiology Spectrum

June 3, 2022

Re: Revised version our manuscript entitled “*Impacts of dietary protein and niacin deficiency on reproduction performance, body growth and gut microbiota of maternal hamsters (Tscherskia triton) and their offspring*” (RE: Spectrum00157-22)

Dear Dr. Theis

Thank you very much for your decision letter and review comments about our manuscript (RE: Spectrum00157-22) on 7 March 2022. The review comments and suggestions are very valuable to us in improving the manuscript. We have carefully considered these comments in revising our manuscript. Point-by-point responses to each of the comments are below the letter. The major revisions are highlighted in red color in the revised manuscript (uploaded as a “Marked Up Manuscript”).

Please let us know if you need us to make further revisions or clarifications.

Warm regards,

Zhibin Zhang, on behalf of all coauthors

Professor

Institute of Zoology, Chinese Academy of Sciences

Beijing, 100101, P.R. China

E-mail: zhangzb@ioz.ac.cn

Appendix: Point-by-point responses to the comments raised by the two reviewers

Reviewer #1:

Zhao *et al.* use an experimental approach to test whether reduced levels of dietary components (protein and niacin) influence aspects of the reproductive biology and diversity of the gut microbiome of greater long-tailed hamsters.

I think that some sections of the manuscript would benefit from re-structuring to improve clarity. This is particularly true for the Introduction. There are a number of issues concerning both bioinformatics and statistical analyses (e.g. the use of cluster instead of error-model -based methods to process sequence variants). In addition, there is a lack of detail regarding data collection. Altogether, these issues make me question the robustness of the results and I suggest that the authors revise the manuscript to improve transparency and clarity. Finally, there are a number of confounding factors that could influence the outcomes of this study. Some of these can be controlled for statistically (i.e., include sex and litter size as explanatory variables in the relevant analyses). But some confounding effects intrinsic to the experimental approach (i.e. legacy effects of housing) are not addressed. The authors should discuss the limitations of their study. I develop these and other concerns and suggestions on how to improve the manuscript in detail bellow. I hope the authors find these comments useful.

Response: Thank you for the valuable comments. We have addressed the issues you raised (For details, see below).

1. 16S rRNA gene raw sequences need to be made available in the NCBI Sequence Read Archive (SRA).

Response: We have submitted our raw sequence data to NCBI SRA under Bioproject: PRJNA816668.

Introduction:

2. A main premise of the study is that nutritional deficiency has a negative effect on

both reproduction and gut microbiome diversity of greater long-tailed hamsters and that such findings support the notion that nutritional deficiency is a major driver of population decline of this species. Indeed, a large section of the introduction focuses on this issue. Whilst I agree that this is a very relevant question, the approach used in this study (e.g., the use of lab-reared animals, punctual sampling of fecal samples, etc.) falls short in its scope to answer it. Long-term monitoring of natural populations or a population level study comparing sites with a gradient of nutritional stressors, for instance, seems more appropriate to evaluate the role of agricultural practices on fitness and whether they contribute to population declines. It seems like most of the introduction is off-topic. Therefore, I would suggest that the authors instead place greater emphasis on developing specific hypothesis regarding experimental manipulation of nutritional factors and their effect on reproduction and host-associated microbiomes.

Response: We have re-written the introduction as suggested.

In the first paragraph, we added a review on the effects of nutrition a deficiency on body growth, reproduction of hosts (line 53-59). In the second paragraph, we added a review on the effects of nutrition deficiency on host-associated gut microbes (line 60-70). In the third paragraph, we modified the review on the effects of nutrition deficiency on wild animals (line 71-80). In the fourth paragraph, we shortened the description of the Greater long-tailed hamster (line 81-90). In the last paragraph, we added a brief introduction about our methods for facilitating reading, and we proposed two hypotheses: (1) Maternal low protein and niacin deficiency would have a negative effect on body growth and reproduction performance of maternal hamsters, and the body growth of offspring, (2) Maternal low protein and niacin deficiency would alter gut microbes of both maternal and offspring hamsters (line 112-119).

Results:

3. L. 111-112: In the context of fitness estimates, body condition represents a more

biologically informative measure than weight (see e.g., Peig & Green 2009 Oikos). Also, please report estimates of repeatability.

Response: Indeed, body length (or tail length) of rodents is often used to calculate the scaled mass index (SMA) as body condition according to the method proposed by Peig & Green (2009, Oikos). Unfortunately, we were not able to measure the body or tail length of Greater long-tailed hamster because the species is solitary and very aggressive. It is hard to measure the body length of living hamsters in the study. To minimize the handling stress, we only measured the body weight of the female hamsters during the diet treatment period. We added a few sentences to explain why we did not use the body or tail length (line 386-389).

We used *rptGaussian* function in *rptR* package to estimate the repeatability. Repeatability values for body weight of female hamsters was calculated by controlling individual identity as a random effect, and nutritional treatments (protein and niacin content diets) as fixed effects. Details of the mixed-effects model were explained in Material and Methods part (line 485-487).

4. Figure 1B: this figure is not clear, what does "reproductive output" represent? Why use proportions?

Response: Sorry for the confusion. We redefined the variables of the figure. In the revised version (Figure 2), we presented the significant effect of low protein diet on body weight of maternal hamsters (Figure 2A) and offspring hamsters (Figure 2B), number of cohabitation tests (Figure 2C), mount latency (Figure 2D), and litter size (Figure 2E). The non-significant effects of niacin diets were shown in Table S3.

5. L. 129: Do "CD" stand for Control treatment? Should it be CON? Is so, please be consistent throughout the text.

Response: We have corrected the abbreviation of different diet treatment group (low protein diet: LPD; normal protein diet: NPD) throughout the manuscript.

6. Line 129-133: Please use a table to report the group-specific (i.e., treatment) estimates (including R²) based on PERMANOVA analysis, not sure what is being reported here. Also, based on the results, I do not think you can argue that groups are "strongly separated" (especially since R² estimates are not reported). Indeed, Figure 2 A does not suggest differentiation at all.

Response: We added a table (Table 2) to report PERMANOVA results including R² estimate. We re-worded the sentences as “We found a significant effect of LPD on the gut microbial community of maternal hamsters during the one-month adaptation to standard and modified AIN93G rodent chow ($p = 0.009$, Table 2, Figure 3A), but no such significant effect of a niacin deficient diet on maternal hamsters ($p = 0.412$, Table 2, Figure 3A). Both maternal LPD ($p = 0.001$, Table 2, Figure 3B) and Niacin⁻ ($p = 0.001$, Table 2, Figure 3B) significantly altered the gut microbial community of offspring at weaning” in line 149-154.

7. PCoA s (Figure 2 and 5): there seem to be very large differences in beta-dispersion between groups (particularly among offspring samples) - this violates a key assumption of PERMANOVA that groups have similar dispersions. The authors should report if there are significant differences in beta-dispersion between groups.

Response: Thank you for this suggestion. We found there was a significant beta dispersion of microbial community between maternal normal protein and low protein offspring ($F = 25.03$, $p = 0.001$). We added the results of beta-dispersion in Table 2 and Table S8 (comparison of predicted functional orthologs and pathways). We also revised the materials and methods in line 508-509.

8. L. 134-141: Did you control for differences between female and male offspring?
The litter sizes?

Response: We added sex (categorical variable) and litter size (continuous variable) as fixed factors of in the models, and no differences between them were found. We reported the results (Line 167-168) and added information in the method (line506-507)

9. L.164-172: As with taxonomic based analysis, please follow the recommendations above for reporting the results of multivariate analysis on functional predictions.

Response: We updated the results and added a table (Table S8 in the revised supplementary material) to report the detailed information.

Materials and Methods

10. Experimental design: How many different genotypes were used in this study? Do all females come from different litters or are some individuals closely related?

Response: All hamsters used in this study were from different litters. We added this information in the method (line 366-367).

11. It is not clear what are the final sample sizes across treatments and how many individuals were used for each analysis. For example, L. 341 "44 females were separately housed.", L. 345 "51 female hamsters were randomly allocated to four diet groups", while Table 1 summarizes results from a subset of 46 individuals. This is quite confusing. A table would make this information more accessible and transparent.

Response: Sorry for this confusion. Sample size changed during the experiment

due to injuries or mortality. We used a table (Table S1) to report change of the sample size due to the following reasons. 6 female hamsters died during the adaptation period, and 3 females displayed aggressive behavior on males and got injured in the cohabitation test. 7 females rejected all males (>10) for mating during the cohabitation test within 14 days, and 16 females were not pregnant after mating with males.

12. Sample collection: (1) Were all fecal samples from females collected at the same time? How old were they? (2) Offspring fecal samples were collected after weaning, but its not clear how old was each individual sampled. Was the timing of weaning different across treatments? Please provide more details. (3) How many individuals from the "offspring" group were female / male? (4) How many individuals were sampled per litter?

Response: We clarified these questions in Figure 1 showing the schematic overview of the experimental design.

(1) Fecal samples from maternal hamsters (aged 6-7 months) were collected within two days (to get enough amount feces for DNA extraction) after a 30-day's diet treatment.

(2) Fecal samples and body weight data of offspring hamsters were obtained immediately at weaning (age 26~27 days), but timing of weaning differed across individuals, because maternal females were not pregnant at same day (Figure 2C).

(3) We added results for litter size and sex ratio in line 141-145 and Table S3.

(4) We obtained a total of 72 pups from four treatment groups (5, 6, 5, and 3 litters from the following diet treatment groups: normal protein & niacin⁺, normal protein & niacin⁻, low-protein & niacin⁺, and low protein & niacin⁻, respectively), 68 pups were sampled for fecal microbiota analysis. We observed one litter of infanticide in niacin deficient diet group (Table S1); pups from this litter all died or seriously harmed by dam within two weeks after birth. We excluded them for microbiota-related analysis. We added the above information in Table S1.

13. Amplicon sequencing: (1) What is the sequencing depth (total and per sample average/range)? (2) How many OTUs were kept after prevalence and abundance thresholds were applied to the data? (3) More importantly, were negative controls (extraction and PCR) used?

Response:

(1) Sequences depth was 50000, feature reads ranged from 9963 ~ 45253, average reads were 22220 per sample. We added this information in line 477-478.

(2) A total of 5791 OTUs (ASVs) was applied for further analysis. We added this information in line 475.

(3) Sterile water was used as negative control. We added this information in Line 464-465.

14. Clustering sequences based on a threshold (OTUs) provide limited resolution. Moreover, clustering (OTU) vs error-model based (ASV) have been shown to influence biological interpretations (e.g. Joos et al. 2020, BMC Genomics; Chiarello et al.. 2022 PlosOne). I would suggest that authors use a more up-to date approach for down-streaming analysis (i.e. dada2).

Response: Thank you very much for the suggestion. We re-run our raw sequence data under qiime2 platform and used DADA2 in quality control and feature table construction. We updated all statistical analysis, tables, and figures. Results of the new analysis are similar as those generated in QIIME pipeline.

15. Please make the 16S rRNA gene raw sequences are made available in the NCBI Sequence Read Archive (SRA)

Response: We have submitted the raw sequences data to NCBI Sequence Read Archive under Bioproject number: PRJNA816668.

Statistical Analysis:

16. L. 395 - 399: (1) Please use generalized linear models (with non-normal errors) instead of Kruskal-Wallis test or a Fisher exact test. (2) In addition, please report estimates of effect size.

Response: Thank you very much for the suggestion.

(1) We re-analyzed reproduction-related behavior data using GLMs, and we found significant difference in mount latency (time latency to the first copulation). We added the results in Figure 2D of the revised version.

(2) We used GLMs to re-analyze the effects of diet treatment on reproduction-related variables. We reported R^2 (calculated with R package *rsq*) as the effect size of GLMs (Table S3).

17. L. 401-403: PcoA is a multivariate method used for exploration and visualization, but it does not allow inference of statistical significance- for this PERMANOVA is commonly used. Also, not sure what "the significance of the beta diversity" mean. I would suggest rewording, something in the lines of "Test for differences in community structure according to nutritional treatment".

Response: Sorry for the unclear description. We re-worded these sentences as follows: "Tests for difference between diet groups in beta diversity were performed using two-way PERMANOVA in *vegan* package (72) with *adonis* function. Ordination analysis was performed using PCoA (Principal Coordinate Analysis) based on Bray-Curtis matrix using *vegan* package in R" (line 509-512).

18. L. 405-418: Was rarefaction applied? How do authors control for sample read depth? This information is crucial, especially when using a method like LefSe.

Response: The minimum sample reads was 9963. All samples were rarefied to 9000 reads. We added the information in line 488-490. In the revised version, we

used ANCOMBC to analyze differential taxa.

19. A major challenge of amplicon-based 16S profiling of bacterial communities is that relative abundance corresponds to compositional data with much debate surrounding which method is more appropriate when evaluating "differential abundance" (also, see Nearing et al. 2022 NatComm for further discussion and implications around this topic). Please use a method that properly address this issue (e.g. Morton et al. 2019 NatComm; Mandal et. al. 2015 Microbiome; Lin & Das Pedada 2020 NatComm).

Response: Thank you very much for the suggestion. We used ANCOM-BC (Lin & Das Pedada 2020 NatComm) algorithm (ANCOMBC package in R) to analyze differential abundance data. We removed results based on random forest. We added (or updated) the results in line 172-191, and method in line 513-519.

Discussion:

20. I think the authors need to be more careful with regard to inferring causality. For instance, in L192-196 "cause disorder of their gut microbiota" and L 194-195 "amplified in offspring via dysfunction of microbiota". With this data, it is not possible to infer dysbiosis (see eg. Hooks & O'Malley 2017 mBio). In this context, I would also urge the authors to soften the title of the manuscript and to not infer causality.

Response: Thank you very for the suggestion. Indeed, it is inappropriate to infer the causality based on our results, although we found significant difference of microbial function pathways between normal and low protein diet groups. In order not to infer the causality, we modified the title as "*Impacts of dietary protein and niacin deficiency on reproduction performance, body growth and gut microbiota of maternal hamsters (Tscherskia triton) and their offspring*". We also deleted statements inferring causality by citing the reference (Hooks & O'Malley 2017

mBio) in discussion (line 268-271, 349-351).

21. There is no discussion of the limitations of this study. A major concern is that even if environmental factors are kept constant in the lab, variation in microbial input (i.e. cage or legacy effects) is likely to be an important (if not the main) factor influencing the findings reported in this study. This should be acknowledged and properly discussed.

Response: We added a paragraph to discuss the limitations of this study (line 336-351).

22. There were a number of grammatical errors throughout the text, including missing punctuation and awkward wording. These are some examples but there are many more in the text:

L. 56: Should be "rodent species are vital" instead of "rodent species is vital";

L. 58: Should be "indiscriminate" instead of "indiscriminative".

L. 60: Should be "facing sharp declines and extirpation" instead of "facing sharp decline or even extinct at"

L. 341, L. 345: Avoid starting a sentence with a number.

L. 362: Should be "In total," instead of "Totally,"

L. 363: Should be "due to death" instead of "due to dead".

L. 370: " fecal samples" instead of " fecal sample."

Response: Thank you very much for pointing out these mistakes. We have corrected them. We also asked a native English speaker to polish the English writing.

Reviewer #2:

The study by Zhao submitted for publication in Microbiology Spectrum aims to understand the transgenerational fitness effects of nutritionally poor diets. The

experiment (which isn't clear early on in the manuscript) contrasts mating behaviour, maternal and offspring gut microbial diversity in groups of hamsters fed with either protein or niacin deficient diets. The authors loosely tie this to threats to rodents in anthropogenically altered landscapes. While I think the study has intrinsic merit and tackles an interesting idea, the ideas, reasoning and presentation of their results are confusing and hard to follow. The experimental design and statistical evaluation are also unclear. As such, I cannot recommend this study to be published without major revisions.

Check grammar, phrasing and spelling throughout. It is worth considering proof-reading by an independent colleague or service.

Response: Thank you very much for these comments and suggestions. We have addressed the issues raised by you (For details, see below)

Abstract:

1. L16: The first two sentences do not link well with one another and do not spell out strongly enough the logical connection between maternal effects, nutrient deficiency and intensified agricultural practices.

Response: We rewrote the introduction to strength the logical connection between maternal effects, nutrient deficiency and intensified agricultural practices. In the first paragraph, we added a review on the effects of nutrition a deficiency on body growth, reproduction of hosts (line 53-59). In the second paragraph, we added a review on the effects of nutrition deficiency on host-associated gut microbes (line 60-70). In the third paragraph, we modified the review on the effects of nutrition deficiency on wild animals (line 71-80). In the fourth paragraph, we shortened the description of the Greater long-tailed hamster (line 81-90). In the last paragraph, we added a brief introduction about our methods for facilitating reading, and we proposed two hypotheses (line 112-116).

2. L24: Replace "Greater long-tailed hamsters" with either "hamsters" or T. triton.

Response: We re-wrote the abstract, and replaced "Greater long-tailed hamsters" with "hamsters" in abstract.

3. L24 and following: (1) There was no background or reasoning as to why microbiota was looked at nor explained when you discuss the study question and design. (2) How did you quantify less responsive?

Response: (1) we re-wrote the abstract (line 17-20) and the introduction (line 60-70) by highlighting the background as to why we look at the gut microbiota.

(2) PERMANOVA tests showed niacin diet showed significant effect on microbial community in offspring, not in maternal hamsters. We modified this sentence in line 33-35.

4. L30: grammar issues. Its "increases" and "decreases"

Response: We corrected this grammar error.

5. L33: nowhere was explained that this study is an experiment rather than a field study. Those details need to be given in the abstract. The links between protein deficient diet, niacin deficiency and monoculture are still obscure even at the end of the abstract.

Response: Our study is a lab experiment. We added the information in the abstract (line 20-23). Because the first reviewer pointed out that the link of nutrition deficiency to hamsters in field, and focus on relationship between nutrition deficiency, fitness and microbes, we re-wrote the introduction, and soften the links to wild populations (line 71-80). We added the need for further investigation on the links of the results in laboratory condition to wild condition by the end of the

abstract (line 35-37), and in IMPORTANCE part (line 47-50).

Importance:

6. L37: the first two sentences here are a much better line of argument than in the abstract.

Response: We have re-written the Abstract and Importance and modified related sentences.

7. L38: I think this sentence is poorly incorporated. There is a logical leap from population declines to maternal effects (which are a very small factor overall contributing to offspring fitness)

Response: We re-wrote the sentence in line 40-41.

8. L42. Grammar issues in this sentence

Response: We corrected the mistakes in this sentence.

Introduction:

9. L50: "making some of them 53 successful living organisms in the earth" - poor word choices and phrasing. L56: "is" = "are".

Response: We reworded this sentence and corrected grammar error.

10. L58: what do you mean with proper? This sentence does make very little sense

Response: We have deleted this sentence.

11. L72: the paragraph ending here should be at least two:1) agricultural

intensification and its impact on biodiversity, 2) rodent diversity and its functional importance + threats from monoculture

Response: We have re-written this paragraph in which the two points have been reflected in line 71-90.

12. L73: poor phrasing; the paragraph feels confused as to what it wants to say; why mention disease here if your main focus is diet? If you want to talk about disease, you can do so in the discussion and suggest that poor fitness and gut microbial diversity can also lead to more severe diseases/infections.

Response: We deleted redundant sentences.

13. L87: this is very confusing. Unfortunately, you have not built a case as to how "maternal dysfunction of gut microbiota" could even affect offspring development.

Response: Indeed, it is improper to use "dysfunction" here. We deleted related phrase and modified our hypothesis in introduction part line 112-116.

14. L102: Don't use apparently.

Response: We deleted this word.

Results:

15. L111: on which response variable. Since you are submitting to a journal with a different, short format that puts the methods at the end, you still need to write the results with as much information as needed to understand what you found.

Response: Thank you for this suggestion. To facilitate reading, we added a brief information of method in introduction to describe our experiment (line 91-112).

16. L115: it's still unclear to me what you done. The last paragraph of the introduction needs to also specify what type of experiment was done etc - there is so little information to go by. Only by looking at the methods section, I can see what you did, but this does not work in a format like this.

Response: We re-wrote the introduction part and added a short description of our experiment design line 91-112.

17. L118: indicate the direction of your effect, i.e., increase/decrease etc.

Response: We used a glm (generalized linear model) to test the effect of diets on litter size. The direction of the effect was added in line 141-142 (also in Figure 2E, Table S3).

18. L122: Is this behaviour in one group meaningful, i.e., did you test for its effect?

Response: We only observed one case of infanticide during pre-weaning period. It's impossible to test its effect. We only discussed it briefly as to if the infanticide was related to nutrition deficiency (line 228-233).

19. Overall, the graphs look nice. Maybe consider polygons for the different diet groups in the PCoA plot.

Response: Thanks for your suggestions. We used circles to cover samples from different diet groups in the ordination plot (see Figure 3A, B; Figure 5A, B).

20. I have to reiterate that the link between maternal fecal microbiome and offspring is not clear: to spell it out, for me, the link is, that differently fed mothers inherit, i.e., seed their offspring with different microbes. But what is not clear to me is

whether the offspring are feeding themselves on maternal milk or also the diet. Can't even get that from the methods.

Response: Thank you for pointing this out. Both maternal and offspring hamsters share several differential taxa (line 187-189), suggesting the links between them, but the detailed pathways from maternal hamsters to offspring hamsters are still unknown. We added a discussion on the link between maternal fecal microbiome and offspring in line 283-289, 342-344.

We did not observe pups eating rodent chow before weaning. Fecal samples from offspring hamsters were immediately collected when they were separated from mother hamsters. We added this information in the method section line 430-432.

21. L164: if neither of the treatments impacted microbial functions, how do you then argue that the treatments have an effect on host fitness?

Response: Indeed, we did not observe significant differences in abundance of functional orthologs and KEGG pathway in maternal hamsters, but we found maternal low-protein diet altered offspring's gut microbes; the function and pathway in low-protein diet groups were significantly from normal protein diet groups. We modified the relevant statements in 192-200.

22. L172: It concerns me that with two comparable functions you end up with two contrasting results.

Response: Sorry for the confusion here. Tax4fun2 package in R could generate two relative abundance tables (predicted functional orthologs and KEGG pathways) by using unique sequence and feature table as input data (Both unique sequence and feature table were generated by QIIME2 (or QIIME in unrevised manuscript)). We first used the PERMANOVA to test the differences of overall

functions and pathway composition between different dietary groups. Second, if there were differences of overall composition between dietary groups, then we performed LEfSe method to obtain differential functional orthologs or KEGG pathways.

In our revised manuscript, we added results of PERMANOVA for tests of the effects of diets on overall composition of functional orthologs and KEGG pathways (Table S8), we observed that maternal LPD significantly changed the composition of both functions ($p = 0.002$) and pathways ($p = 0.016$) in offspring hamsters, we then performed LEfSe analysis to obtain differential pathways (Table S9). In our study, we only discussed the differential KEGG pathway and did not discuss the differential functional orthologs.

Methods:

23. There is too little information on the microbiome sequencing and data processing (e.g., reads, filtering parameters)

Response: We added more detailed information on sequencing, reads rarefied, quality control etc. in the method section (line 451-465, 466-482). Besides, we updated methods in amplicon sequencing (QIIME2 platform and dada2 denoise method) as suggested by the first reviewer.

24. Is this QIIME or QIIME2 - there are differences between the versions and I would recommend the newer one.

Response: Thanks for your recommendation, we re-run our data using QIIME2(2022-2), and updated all results, including Tables and Figures. Results of new analysis on diversity differences (alpha diversity and microbial community) were similar with those generated in QIIME pipeline.

25. Why where some stat test calculated in QIIME and others in R. The reporting of

the statistical test is hard to follow.

Response: We updated the amplicon analysis, all statistical analysis were performed in R (including LMM, GLMs, anova, betadisper, PERMANOVA, ANCOMBC) except LefSe (<http://huttenhower.sph.harvard.edu/galaxy/>).

We added the details on package or functions from R in Statistical analysis section of Materials and methods.

We re-wrote the statistical part to make a clear presentation on models we used.

26. L390: Grammer.

Response: We corrected the grammar error with help of a native English speaker.

August 7, 2022

Prof. Zhibin Zhang
Institute of Zoology, Chinese Academy of Sciences
Beijing
China

Re: Spectrum00157-22R1 (Impacts of dietary protein and niacin deficiency on reproduction performance, body growth and gut microbiota of maternal hamsters (*Tscherskia triton*) and their offspring)

Dear Prof. Zhibin Zhang:

Thank you for submitting your revised manuscript to Microbiology Spectrum. The manuscript has been reconsidered by two of the prior reviewers. I do apologize for the delayed decision. While both have indicated that the manuscript is much improved, and is therefore close to publication, three primary concerns remain. First, further grammatical editing of the manuscript is required. Second, when conducting PERMANOVA, homogeneity of dispersion, or a lack thereof, needs to be considered as an explanation for potential significant effects (See comments by Reviewer 2). Also, please address the issue of accounting for dams' IDs. Third, please address Reviewer 1's concerns about rarefaction versus treating microbiome data as compositional. I recognize that this is an active debate, and if you choose the rarefaction approach and explain why you have done so, that will be considered. Yet, if ideally you are able to demonstrate that this does not affect the principal results, then the current framework can remain and a statement about lack of influence can be placed within the Methods.

When submitting the revised version of your paper, please provide (1) point-by-point responses to the issues raised by the reviewers as file type "Response to Reviewers," not in your cover letter, and (2) a PDF file that indicates the changes from the original submission (by highlighting or underlining the changes) as file type "Marked Up Manuscript - For Review Only". Please use this link to submit your revised manuscript. Detailed instructions on submitting your revised paper are below.

Link Not Available

Sincerely,

Kevin R. Theis

Journals Department
Reviewer comments:

Reviewer #1 (Comments for the Author):

Overall, I appreciate the efforts made by the authors to revise the manuscript according to the previous reviews. In particular, there is more clarity regarding data collection, data analysis has been revised and more up to date methods were implemented.

Also, the introduction and discussion sections are easier to follow.

However, I have some major concerns:

1. There are still grammar issues and poor choice of wording. For instance "maternal hamsters" should be "female hamsters" or "mother hamsters". Other examples:

L. 300-303. "Genus *Treponema* (3.00% in relative abundance of all fecal samples), *Desulfovibrio* (2.54%), *Sporobacter* (0.30%), *Helicobacter* (0.30%), *Dorea* (0.21%), and *Ruminococcus* (0.21%) were the top predominant taxa in all samples based on the relative abundance." I struggle to understand what is being said here.

L 304-306. Also not sure what do the authors mean by "abundance of *Desulfovibrio* is mediated by many other bacteria groups, which may lead to inconsistencies or even contradictions between studies"

L 415. "to represent the body growth conditions of offspring hamsters." - Perhaps just refer to body weight instead of body growth.

L417. "recorded" instead of "calculated".

These are some examples but there are more in the text.

2. Although for the most part the authors avoid inferring causality, some of the author's interpretations go beyond their results. For instance:

- L 31- 33. " Our results suggest that low-protein and niacin deficiency may result in negative impacts on the fitness of animals through altering their gut microbiota." The authors find a relationship between diet treatments, body weight, reproductive behaviour, and some aspects of gut microbial diversity. The experimental approach is not designed to test the indirect effect of the microbiome on variation in reproductive behaviour and body weight.

- L 116-119. The authors experimentally manipulate the diet of hamsters under lab conditions to evaluate whether a deficiency of protein or niacin has an effect of reproductive output, weight and the diversity of the gut microbiome. As mentioned in the previous round of reviews, claiming that a significant relationship between these variables provides evidence of the role of nutrition on population declines is too far fetched. What can be said is that it might have important implications for natural populations and more work needs to be done to confirm this potential issue.

3. The authors made an impressive effort to improve clarity as well as implementing more appropriate analysis. A few remaining concerns:

- Rarefaction raises several concerns. As an alternative, sequencing depth can be included as a model covariate to control for differences in library size.
- 16S amplicon data is compositional and should be treated as such. For differential abundance, the authors use ANCOM-bc, a reliable method in the context of compositional data. Following this approach, I suggest to also use compositional data analyses for clustering (*PcoA*) (see e.g. Quinn et al. *GigaScience*, 2017).
- For ANCOM-bc analysis, please clearly state the formula used (not sure if covariates were included). Also, which procedure was used to correct for multiple testing? What was the significance threshold used to define false discovery rate? How many ASV's were included in this analysis (i.e. after applying a filter of `zero_cut=0.9`)? I think the results of this analyses should be integrated in the main text as a figure rather than supplementary information. Please also report confidence intervals.

4. Overall, I think the authors need to avoid the use of overly generic statements when discussing their findings. For example, L. 312-317 or L. 349-351. Also, when addressing the limitations of the study, I suggest to discuss their implication when interpreting results, rather than compiling a list of caveats and how they need to be incorporated in future studies.

Reviewer #2 (Comments for the Author):

This is my second time reviewing Zhao et al.'s manuscript and it is much improved.

Abstract:

Major: The abstract does not point out the interesting results. The fact that diet affects host microbiota is well known, but that it impacts offspring (possibly via lactation, although never mechanistically explained) is interesting.

L25 and following: I would suggest to call the adult hamster mothers "hamster females" rather than "maternal hamsters" throughout the manuscript.

L30: bacterial instead of bacterium

L31: delete first "in offspring"

L34 and following: its either "nutritional deficiency" or "nutrient deficiency", but not nutrition deficiency

L34 and following: use "protein deficient group" or "low protein group" rather than "protein deficiency group"

Introduction:

The first paragraph is weak, but the structure of the rest is fine and at times well written. In other places, the introduction would require some editing by a native English speaker.

To improve the first paragraph, I would recommend to the authors to consider following questions: Why is a nutritional balanced diet important? How do nutrients impact organismal homeostasis and individual fitness? What nutrients are important and why is protein particularly relevant? What is Niacin and why is it also important? What relationship has a niacin rich diet with a protein rich diet? (some of the information you detail in your discussion L327-330). Now to link this with the wider consequences of human changing nature, you might also want to consider, what causes changes to the nutritional profile of food for wildlife?

Table 1. Give each diet (NPD-Niacin+; NPD-Niacin-; LPD-Niacin+; LPD-Niacin-) a separate column and bolden the differences in protein and Niacin content.

Betadisper tests homogeneity of dispersion among groups (regions in your case), which is a condition/assumption of the PERMANOVA. The latter tests no difference in 'location', that is, tests whether composition among groups is similar or not. You may have the centroids of two groups in NMS at a very similar position in the ordination space, but if their dispersions are quite different, adonis will give you a significant p-value, thus, the result is heavily influenced not by the difference in composition between groups but by differences in composition within groups (heterogeneous dispersion, and thus a measure of betadiversity). The difference in dispersion in your data could thus also be the reason for your significant result in the PERMANOVA. So for the interpretation of this result one must consider that it is ideal to have equal sample sizes for both analyses. For an unbalanced experimental design this can either increase rejection rates or the PERMANOVA can become more conservative. This is likely the case here (already very small litter sample size of 5,5,5, and 3 for each of the 4 diets; sample size for female hamsters on each diet not included as stated in Table 1 - see L389). Include sample size either in table 1 or/and Figure 1. I would also recommend the authors to consider choosing an extension to Permanova to deal with unequal sample sizes (see advise on unbalanced designs in section 4:

<https://onlinelibrary.wiley.com/doi/full/10.1002/9781118445112.stat07841>)

This also makes me wonder whether for the PERMANOVA analysis on offspring microbiota the mother's id should be a random effect to account for variation between these individuals.

Discussion:

The authors managed to discuss many of the limitations of the study. Well done.

The authors can speculate in the discussion a bit more of how their work relates to what we expect nutrient deficient diet to do in nature. Conservation concerns can be raised here.

Overall, the authors managed to incorporate the feedback successfully.

I am still concerned with some of the writing. The MS is wordy, uses clunky language and sentence structures and often wording is off, but it is also worth mentioning that its much better than in the original version.

Staff Comments:

Preparing Revision Guidelines

Please return the manuscript within 60 days; if you cannot complete the modification within this time period, please contact me. If you do not wish to modify the manuscript and prefer to submit it to another journal, please notify me of your decision immediately so that the manuscript may be formally withdrawn from consideration by Microbiology Spectrum.

This is my second time reviewing Zhao et al.'s manuscript and it is much improved.

Abstract:

Major: The abstract does not point out the interesting results. The fact that diet affects host microbiota is well known, but that it impacts offspring (possibly via lactation, although never mechanistically explained) is interesting.

L25 and following: I would suggest to call the adult hamster mothers "hamster females" rather than "maternal hamsters" throughout the manuscript.

L30: bacterial instead of bacterium

L31: delete first "in offspring"

L34 and following: its either "nutritional deficiency" or "nutrient deficiency", but not nutrition deficiency

L34 and following: use "protein deficient group" or "low protein group" rather than "protein deficiency group"

Introduction:

The first paragraph is weak, but the structure of the rest is fine and at times well written. In other places, the introduction would require some editing by a native English speaker.

To improve the first paragraph, I would recommend to the authors to consider following questions: Why is a nutritional balanced diet important? How do nutrients impact organismal homeostasis and individual fitness? What nutrients are important and why is protein particularly relevant? What is Niacin and why is it also important? What relationship has a niacin rich diet with a protein rich diet? (some of the information you detail in your discussion L327-330). Now to link this with the wider consequences of human changing nature, you might also want to consider, what causes changes to the nutritional profile of food for wildlife?

Table 1. Give each diet (NPD-Niacin+; NPD-Niacin-; LPD-Niacin+; LPD-Niacin-) a separate column and bolden the differences in protein and Niacin content.

Betadisper tests homogeneity of dispersion among groups (regions in your case), which is a condition/assumption of the PERMANOVA. The latter tests no difference in 'location', that is, tests whether composition among groups is similar or not. You may have the centroids of two groups in NMS at a very similar position in the ordination space, but if their dispersions are quite different, adonis will give you a significant p-value, thus, the result is heavily influenced not by the difference in composition between groups but by differences in composition within groups (heterogeneous dispersion, and thus a measure of betadiversity). The difference in dispersion in your data could thus also be the reason for your significant result in the PERMANOVA. So for the interpretation of this result one must consider that it is ideal to have equal sample sizes for both analyses. For an unbalanced experimental design this can either increase rejection rates or the PERMANOVA can become more conservative. This is likely the case here (already very small litter sample size of 5,5,5, and 3 for each of the 4 diets; sample size for female hamsters on each diet not included as stated in Table 1 – see L389). Include sample size either in table 1 or/and Figure 1. I would also recommend the

authors to consider choosing an extension to Permanova to deal with unequal sample sizes (see advise on unbalanced designs in section 4:
<https://onlinelibrary.wiley.com/doi/full/10.1002/9781118445112.stat07841>)

This also makes me wonder whether for the PERMANOVA analysis on offspring microbiota the mother's id should be a random effect to account for variation between these individuals.

Discussion:

The authors managed to discuss many of the limitations of the study. Well done.

The authors can speculate in the discussion a bit more of how their work relates to what we expect nutrient deficient diet to do in nature. Conservation concerns can be raised here.

Overall, the authors managed to incorporate the feedback successfully.

I am still concerned with some of the writing. The MS is wordy, uses clunky language and sentence structures and often wording is off, but it is also worth mentioning that its much better than in the original version.

Microbiology Spectrum

August 7, 2022

Re: Revised version our manuscript entitled “*Impacts of dietary protein and niacin deficiency on reproduction performance, body growth and gut microbiota of female hamsters (Tscherskia triton) and their offspring*” (RE: Spectrum00157-22R1)

Dear Dr. Theis

Thank you very much for your decision letter and the reviewer’s comments about our revised manuscript (RE: Spectrum00157-22R1) on 7 August 2022. We are grateful to you and the reviewers for their valuable comments which are very valuable for us to improve the manuscript. We have addressed all of these comments and suggestions. Point-by-point responses to each of the comments are attached below the letter. Major revisions are highlighted in red in the revised manuscript (uploaded as a “Marked Up Manuscript”).

Please let us know if you need us to make further revisions or clarifications.

Warm regards,

Zhibin Zhang, on behalf of all coauthors

Professor

Institute of Zoology, Chinese Academy of Sciences

Beijing, 100101, P.R. China

E-mail: zhangzb@ioz.ac.cn

Appendix: Point-by-point responses to the comments raised by the two reviewers

Reviewer #1:

Overall, I appreciate the efforts made by the authors to revise the manuscript according to the previous reviews. In particular, there is more clarity regarding data collection, data analysis has been revised and more up to date methods were implemented. Also, the introduction and discussion sections are easier to follow. However, I have some major concerns:

Response: Thank you for these valuable comments and suggestions. We have addressed your further concerns and comments (see below).

1. There are still grammar issues and poor choice of wording. These are some examples but there are more in the text.

1). "Maternal hamsters" should be "female hamsters" or "mother hamsters".

Response: We changed “maternal hamsters” to “female hamsters” throughout the manuscript.

2). L300-303. "Genus *Treponema* (3.00% in relative abundance of all fecal samples), *Desulfovibrio* (2.54%), *Sporobacter* (0.30%), *Helicobacter* (0.30%), *Dorea* (0.21%), and *Ruminococcus* (0.21%) were the top predominant taxa in all samples based on the relative abundance." I struggle to understand what is being said here.

Response: Sorry for the confusion. We rewrote these sentences as follows “We found abundance of the top six taxa, including *Treponema*, *Desulfovibrio*, *Sporobacter*, *Helicobacter*, *Dorea*, and *Ruminococcus* were decreased, but probiotics were enriched in offspring of maternal LPD diet group (Figure 5C;

Table S6). No such effects were observed in offspring of maternal Niacin⁻ diet group. These observations suggested that gut microbiota of hamsters was more sensitive to dietary protein deficiency than to niacin deficiency” in line 301-306.

3). L304-306. Also, not sure what do the authors mean by "abundance of *Desulfovibrio* is mediated by many other bacteria groups, which may lead to inconsistencies or even contradictions between studies".

Response: We re-wrote these sentences as follows “Lower abundance of *Desulfovibrio* was correlated with lower BMI (Body Mass Index) (52) and birth weight (53) in humans, which is consistent with our observation that lower body weight of offspring was also associated with lower abundance of *Desulfovibrio*” in line 306-309.

4). L 415. "To represent the body growth conditions of offspring hamsters." - Perhaps just refer to body weight instead of body growth.

Response: We corrected this sentence in line 421.

5). L417. "recorded" instead of "calculated".

Response: We corrected this word.

2. Although for the most part the authors avoid inferring causality, some of the author's interpretations go beyond their results. For instance:

1). L 31- 33. " Our results suggest that low-protein and niacin deficiency may result in negative impacts on the fitness of animals through altering their gut microbiota." The authors find a relationship between diet treatments, body weight,

reproductive behaviour, and some aspects of gut microbial diversity. The experimental approach is not designed to test the indirect effect of the microbiome on variation in reproductive behaviour and body weight.

Response: Thank you for your suggestions. We modified this sentence as “Our results suggest that a low-protein diet could alter gut microbiota in animals, which may result in negative impacts on their fitness” in line 32-34.

2). L 116-119. The authors experimentally manipulate the diet of hamsters under lab conditions to evaluate whether a deficiency of protein or niacin has an effect of reproductive output, weight and the diversity of the gut microbiome. As mentioned in the previous round of reviews, claiming that a significant relationship between these variables provides evidence of the role of nutrition on population declines is too farfetched. What can be said is that it might have important implications for natural populations and more work needs to be done to confirm this potential issue.

Response: Thank you for your suggestion. We modified this sentence as follows “If these hypotheses held, it would provide us with important implications that nutritional deficiency may play a potential role on population decline of *T. triton* under the intensified monoculture in the North China Plain, and more work needs to be done to confirm this potential issue” in line 122-125.

3. The authors made an impressive effort to improve clarity as well as implementing more appropriate analysis. A few remaining concerns:
 - 1). Rarefaction raises several concerns. As an alternative, sequencing depth can be included as a model covariate to control for differences in library size.

Response: Thanks for your suggestions. The original analysis of rarefaction

was conducted by “Moving Picture” tutorial in QIIME2 docs (<https://docs.qiime2.org/2022.8/tutorials/moving-pictures/>) in step “Alpha and beta diversity analysis”. Subsampling is a mandatory argument here; commands were listed as follows:

```
qiime diversity core-metrics-phylogenetic \  
  --i-phylogeny rooted-tree.qza \  
  --i-table table.qza \  
  --p-sampling-depth 9000 \  
  --m-metadata-file sample-metadata.tsv \  
  --output-dir core-metrics-results
```

As you recommended, we re-analyzed alpha diversity indices without subsampling by using *diversity* plugin (command: *qiime diversity alpha*) in qiime2. Then we used the library size as covariate in ancova, and we found the results on significance tests were generally similar to those of our original version, excepted for the Shannon index between different niacin levels (change from nonsignificant to significant).

Based on new analysis as you suggested, we updated the results of alpha diversity in Figure 3C, D and Table S4, and updated the MATERIAL AND METHODS part in line 483-485 and 512-516.

2). 16S amplicon data is compositional and should be treated as such. For differential abundance, the authors use ANCOM-bc, a reliable method in the context of compositional data. Following this approach, I suggest to also use compositional data analyses for clustering (PcoA) (see e.g. Quinn et al. GigaScience, 2017).

Response: Thank you for your suggestion. We re-conducted analysis of clustering. First, zeros in ASVs table were replacing by a Bayesian-

multiplicative replacement strategy using *zCompositions* package in R; then, compositional data was clr-transformed by *composition* package; finally, the Euclidean distance (Aitchison distance) was computed for PCoA. We updated Figure 3A, B, added details of clustering and related references in MATERIAL AND METHODS part in line 517-529.

3). For ANCOM-bc analysis, please clearly state the formula used (not sure if covariates were included). Also, which procedure was used to correct for multiple testing? What was the significance threshold used to define false discovery rate? How many ASV's were included in this analysis (i.e., after applying a filter of zero_cut=0.9)? I think the results of this analyses should be integrated in the main text as a figure rather than supplementary information. Please also report confidence intervals.

Response: The ANCOM-BC analysis was performed using *ancombc()* function in ANCOMBC package with several arguments including phyloseq object (generated by phyloseq package in R, it contained feature table, taxonomic table, metadata, and phylogenetic tree of represent sequence), comparison group (protein or niacin group in our study), p value correction method (we used default method “holm” correction) , cut value, and level of significance (we used default value: 0.05). No covariates were included.

Function *ancombc()* was ran as:

```
ancombc(phyloseq = genus.taxa,  
        formula = "protein + niacin",  
        p_adj_method = "holm",  
        zero_cut = 0.90,  
        lib_cut = 1000,  
        group = "protein", ## or "niacin"  
        struc_zero = TRUE,
```

```
neg_lb = TRUE,  
tol = 1e-5,  
max_iter = 100,  
conserve = TRUE,  
alpha = 0.05, ##  
global = TRUE)
```

We have added these details of these arguments in MATERIAL AND METHODS part in line 534-538.

We had 533 and 737 ASVs left after a “0.9 zero cut” filtering in female and offspring hamster group, respectively.

We added a new figure (Figure 5) to show the results of ANCOMBC by using beta value, i.e., log fold change of differential abundance at genus level.

We added beta value (effect size of ancombc, equal to $W * \text{standard error}$) and its 95% confidence intervals in Table S5, S6, and S7.

4. Overall, I think the authors need to avoid the use of overly generic statements when discussing their findings. For example, L. 312-317 or L. 349-351. Also, when addressing the limitations of the study, I suggest to discuss their implication when interpreting results, rather than compiling a list of caveats and how they need to be incorporated in future studies.

Response: Thank you for your suggestion. We modified the generic statements as follows “Therefore, reduction of these important SCFAs-producing microbes in the LPD group may impose negative effects on both female hamsters and their offspring” in line 317-318. We also re-arranged the paragraph in line 301-318.

We modified the sentence in line 349-351 as follows “Finally, microbiota transplantation experiments and multi-omics analysis are necessary to identify malnutrition-induced gut microbiome disturbances, and to examine their potential impacts on physiological and reproductive performance of hamsters”. (Line 351-354)

Reviewer #2:

This is my second time reviewing Zhao et al.'s manuscript and it is much improved.

Abstract: The abstract does not point out the interesting results. The fact that diet affects host microbiota is well known, but that it impacts offspring (possibly via lactation, although never mechanistically explained) is interesting.

Response: Thank you for your valuable comments and suggestions. We modified the abstract by highlighting the maternal nutritional effect on gut microbes in offspring in line 26-30.

1. L25 and following: I would suggest to call the adult hamster mothers "hamster females" rather than "maternal hamsters" throughout the manuscript.

Response: Thank you for your suggestion, a similar suggestion by the first reviewer. We have changed “maternal hamsters” into “female hamsters” in the text.

2. L30: bacterial instead of bacterium.
- 3.

Response: We corrected this mistake.

4. L31: delete first "in offspring"

Response: We corrected it.

5. L34 and following: its either "nutritional deficiency" or "nutrient deficiency", but not nutrition deficiency.

Response: We correct the phrase and used “nutritional deficiency” here.

6. L34 and following: use "protein deficient group" or "low protein group" rather than "protein deficiency group".

Response: We corrected it.

Introduction: The first paragraph is weak, but the structure of the rest is fine and at times well written. In other places, the introduction would require some editing by a native English speaker.

Response: We have asked a native English speaker to polish the English of the manuscript.

7. To improve the first paragraph, I would recommend to the authors to consider following questions: Why is a nutritional balanced diet important? How do nutrients impact organismal homeostasis and individual fitness? What nutrients are important and why is protein particularly relevant? What is Niacin and why is it also important? What relationship has a niacin rich diet with a protein rich diet? (Some of the information you detail in your discussion L327-330). Now to link this with the wider consequences of human changing nature, you might also want to consider, what causes changes to the nutritional profile of food for wildlife?

Response: Thank you for the suggestions. We have improved the first paragraph by considering these important points. (Line 52-65)

8. Table 1. Give each diet (NPD-Niacin+; NPD-Niacin-; LPD-Niacin+; LPD-Niacin-) a separate column and bolden the differences in protein and Niacin content.

Response: We revised Table 1 as suggested.

9. Betadisper tests homogeneity of dispersion among groups (regions in your case), which is a condition/assumption of the PERMANOVA. The latter tests no difference in 'location', that is, tests whether composition among groups is similar or not. You may have the centroids of two groups in NMS at a very similar position in the ordination space, but if their dispersions are quite different, adonis will give you a significant p-value, thus, the result is heavily influenced not by the difference in composition between groups but by differences in composition within groups (heterogeneous dispersion, and thus a measure of beta diversity). The difference in dispersion in your data could thus also be the reason for your significant result in the PERMANOVA. So, for the interpretation of this result once must consider that it is ideal to have equal sample sizes for both analyses. For an unbalanced experimental design this can either increase rejection rates or the PERMANOVA can become more conservative. This is likely the case here (already very small litter sample size of 5,5,5, and 3 for each of the 4 diets; sample size for female hamsters on each diet not included as stated in Table 1 - see L389). Include sample size either in table 1 or/and Figure 1. I would also recommend the authors to consider choosing an extension to Permanova to deal with unequal sample sizes (see advise on unbalanced designs in section 4: <https://onlinelibrary.wiley.com/doi/full/10.1002/9781118445112.stat07841>).

Response: Thank you for your suggestion. We re-conducted PERMANOVA analysis using PRIMER (version 7.0) software that fit the unbalanced design and enable us to include random factor. We conducted PERMANOVA using compositional data analysis guide as recommended by another reviewer. First,

zero counts in ASVs table were replaced by a Bayesian-multiplicative replacement strategy using *zCompositions* package in R; then, compositional data was clr-transformed by *composition* package; finally, the Euclidean distance (Aitchison distance) was computed for PERMANOVA. However, the predicted KEGG functional orthologs and pathway profiles were relative abundance data, thus, the implementation of PERMANOVA were still based on Bray-Curtis dissimilarity matrix. We revised the related methods in line 517-529.

We also added related reference as follows:

- (1) Quinn TP, Erb I, Gloor G, Notredame C, Richardson MF, Crowley TM. 2019. A field guide for the compositional analysis of any-omics data. *Gigascience* 8
- (2) Clarke K, Gorley R. 2015. Getting started with PRIMER v7. PRIMER-E: Plymouth, Plymouth Marine Laboratory 20.
- (3) Anderson M, Gorley RN, Clarke RK. 2008. Permanova+ for primer: Guide to software and statistical methods. Primer-E Limited.

Based on this new analysis, we updated results of PERMANOVA of both female and offspring hamsters in Table 2 and Table S8. There was no difference between the updated and previous results when using PRIMER software and Euclidean distance for conducting PERMANOVA. However, the results were changed when the random factor was considered.

10. This also makes me wonder whether for the PERMANOVA analysis on offspring microbiota the mother's id should be a random effect to account for variation between these individuals.

Response: Thank you for your suggestion. We re-conducted PERMANOVA of both fecal microbial community structure and predicted function and pathway with random effect.

In fecal microbial community, we found that the p-value of “Niacin” term (p value: 0.001 → 0.207) and “Protein × Niacin” interaction term (p value: 0.015 → 0.584) were changed from significant to non-significant when the dam’ ID was considered as a random factor. The results were updated in Table 2.

Also, in predicted KEGG pathway of offspring hamster, we found the p-value of “Protein” term (0.016 → 0.073) was also changed from significant to non-significant when the random factor was incorporated into PERMANOVA analysis. But the p-value of structure of KEGG orthologs was still significant (0.002 → 0.021). The results were updated in Table S8. We also added a table (Table S10) showing the significantly enriched or depleted KEGG functional orthologs between maternal NPD and LPD offspring.

In general, main results still hold even we include the random factor of dam’ ID.

Discussion: The authors managed to discuss many of the limitations of the study. Well done.

11. The authors can speculate in the discussion a bit more of how their work relates to what we expect nutrient deficient diet to do in nature. Conservation concerns can be raised here.

Response: We added a few sentences of discussion about conservation concerns (Line 361-364).

12. Overall, the authors managed to incorporate the feedback successfully. I am still concerned with some of the writing. The MS is wordy, uses clunky language and sentence structures and often wording is off, but it is also worth mentioning that it’s much better than in the original version.

Response: We have asked a native English speaker to edit this manuscript again.

October 10, 2022

Prof. Zhibin Zhang
Institute of Zoology, Chinese Academy of Sciences
Beijing
China

Re: Spectrum00157-22R2 (Impacts of dietary protein and niacin deficiency on reproduction performance, body growth and gut microbiota of female hamsters (*Tscherskia Triton*) and their offspring)

Dear Prof. Zhibin Zhang:

Your manuscript has been accepted, and I am forwarding it to the ASM Journals Department for publication. You will be notified when your proofs are ready to be viewed. Thank you for submitting your paper to Spectrum.

Sincerely,

Kevin Theis
Editor, Microbiology Spectrum
